# CryoSplat: Gaussian Splatting for Cryo-EM Homogeneous Reconstruction

**Suyi Chen**
Department of Computer Science
Stony Brook University
suychen@cs.stonybrook.edu

**Haibin Ling**[*]
Department of Artificial Intelligence
Westlake University
linghaibin@westlake.edu.cn

## Abstract

As a critical modality for structural biology, cryogenic electron microscopy (cryo-EM) facilitates the determination of macromolecular structures at near-atomic resolution. The core computational task in single-particle cryo-EM is to reconstruct the 3D electrostatic potential of a molecule from noisy 2D projections acquired at unknown orientations. Gaussian mixture models (GMMs) provide a continuous, compact, and physically interpretable representation for molecular density and have recently gained interest in cryo-EM reconstruction. However, existing methods rely on external consensus maps or atomic models for initialization, limiting their use in self-contained pipelines. In parallel, differentiable rendering techniques such as Gaussian splatting have demonstrated remarkable scalability and efficiency for volumetric representations, suggesting a natural fit for GMM-based cryo-EM reconstruction. However, off-the-shelf Gaussian splatting methods are designed for photorealistic view synthesis and remain incompatible with cryo-EM due to mismatches in the image formation physics, reconstruction objectives, and coordinate systems. Addressing these issues, we propose **cryoSplat**, a GMM-based method that integrates Gaussian splatting with the physics of cryo-EM image formation. In particular, we develop an orthogonal projection-aware Gaussian splatting, with adaptations such as a view-dependent normalization term and FFT-aligned coordinate system tailored for cryo-EM imaging. These innovations enable stable and efficient homogeneous reconstruction directly from raw cryo-EM particle images using random initialization. Experimental results on real datasets validate the effectiveness and robustness of cryoSplat over representative baselines. The code will be released at https://github.com/Chen-Suyi/cryosplat.

## 1 Introduction

Single particle cryogenic electron microscopy (cryo-EM) has emerged as a transformative tool in structural biology, enabling visualization of macromolecular complexes at atomic or near-atomic resolution without crystallization (Kühlbrandt, 2014; Nogales, 2016; Renaud et al., 2018). Central to cryo-EM is the computational task of reconstructing a 3D volume from a large collection of 2D projection images, each corresponding to a different, unknown viewing direction of identical particles embedded in vitreous ice.

This inverse problem is inherently ill-posed and computationally challenging. First, cryo-EM images are severely corrupted by noise due to the low electron dose required to prevent radiation damage. For experimental datasets, the signal-to-noise (SNR) could be as low as around $-20$ dB (Bendory et al., 2020; Bepler et al., 2020). Second, the orientations (poses) of individual particles are unknown and must be inferred jointly with the 3D structure. Third, many biological samples exhibit structural heterogeneity, with multiple conformational states coexisting in the same dataset.

These difficulties underscore two central objectives in cryo-EM image analysis: *ab initio* reconstruction, which aims to estimate both the 3D structure and particle poses directly from raw data

---

[*]Corresponding author.

without prior models, and heterogeneous reconstruction, which seeks to disentangle and reconstruct multiple structural states from the dataset. Both objectives fundamentally rely on the availability of a robust and efficient homogeneous reconstruction method, which assumes all particles correspond to a single structure and serves as a building block for more complex inference.

Approaches to homogeneous reconstruction include methods based on backprojection, iterative expectation-maximization with voxel-based volumes (Tang et al., 2007; Scheres, 2012; Punjani et al., 2017; Shekarforoush et al., 2024), and more recently, neural representation learning (Zhong et al., 2021a;b), which models the 3D volume using coordinate-based networks. In parallel, Gaussian mixture models (GMMs) have received attention for their continuous, compact, and physically interpretable parameterization of molecular density (Chen & Ludtke, 2021; Chen et al., 2023a). Notably, GMMs offer a natural connection to atomic models and can represent fine structural details using fewer parameters (Chen et al., 2023b; Li et al., 2024; Schwab et al., 2024; Chen, 2025).

Despite their conceptual appeal, existing GMM-based methods (Chen & Ludtke, 2021; Chen et al., 2023a;b; Li et al., 2024; Schwab et al., 2024; Chen, 2025) for cryo-EM reconstruction rely on non-trivial prerequisite steps. They typically rely on consensus volumes from external pipelines, or even atomic models, for initialization, and have not demonstrated stable convergence when directly optimizing from experimental images. In fact, no prior method achieves reliable GMM-based reconstruction even under known particle poses, due to the inherent difficulty of optimizing mixture parameters in extreme noise. As a result, GMMs lack a self-contained and stable formulation that can serve as a backbone for broader reconstruction workflows.

In this work, we propose **cryoSplat**, a GMM-based homogeneous reconstruction method that fills this foundational gap. Given known particle poses, cryoSplat performs stable and efficient reconstruction directly from raw cryo-EM projection images, starting from random initialization and requiring no external priors. Inspired by recent advances in 3D Gaussian Splatting (3DGS, Kerbl et al. (2023)), we model the 3D density as a mixture of anisotropic Gaussians and project them into 2D using a novel differentiable orthographic splatting algorithm consistent with cryo-EM physics. To support practical scalability and training efficiency, we develop a CUDA-accelerated real-space renderer that enables fast rasterization and optimization of the GMM.

Our contributions can be summarized as follows:

- A self-contained GMM-based reconstruction method: We present cryoSplat as the first method capable of performing cryo-EM homogeneous reconstruction from a randomly initialized Gaussian mixture model without an external prior, thereby establishing the missing foundation needed to develop GMMs into standalone reconstruction tools.

- A physically accurate projection model: We design a splatting algorithm under orthogonal projection tailored to cryo-EM image formation, enabling differentiable projection of anisotropic Gaussians in real space.

- An efficient implementation: We adapt the CUDA tile-based framework of 3DGS to cryo-EM imaging, introducing modified forward equations and re-derived gradients, which enables fast optimization of GMMs with tens of thousands of Gaussians.

- Experimental validation: We demonstrate the effectiveness of cryoSplat on real datasets, showing that it converges reliably from random initialization and achieves reconstruction quality outperforming state-of-the-art methods.

## 2 RELATED WORK

### 2.1 VOLUME REPRESENTATION IN CRYO-EM

In cryo-EM experiments, purified biomolecules are rapidly frozen in a thin layer of vitreous ice, where each particle adopts a random orientation. A high-energy electron beam passes through the specimen, interacts with the electrostatic potential of the particles, and is recorded on a detector as a 2D projection image (Singer & Sigworth, 2020). The goal of cryo-EM reconstruction is to recover the 3D electrostatic potential, i.e., the volume, from a large set of such noisy and randomly oriented 2D projections. Central to cryo-EM reconstruction is the choice of volume representation.

### 2.1.1 VOXEL-BASED REPRESENTATION

Voxel-based representations are the most widely used in conventional cryo-EM software, e.g., RE-LION (Scheres, 2016b), cryoSPARC (Punjani et al., 2017) and EMAN2 (Tang et al., 2007). The 3D volume is discretized into a regular grid of density values, enabling fast projection and reconstruction via FFT-based algorithms. Despite their practical success, voxel grids are memory-intensive, which limits their compatibility with modern learning-based analysis frameworks.

### 2.1.2 NEURAL FIELD

Neural fields represent the volume as a continuous function parameterized by neural networks. These methods (Zhong et al., 2021a;b; Levy et al., 2022a;b; 2025) offer differentiability, implicit smoothness, and natural compatibility with learning-based heterogeneous analysis. However, the implicit nature of neural fields often comes at the cost of interpretability, and such models are typically slow to train and difficult to constrain with biological priors.

### 2.1.3 GAUSSIAN MIXTURE MODEL

Gaussian mixture models have a long history in structural biology, with early uses for molecular approximation (Grant & Pickup, 1995; Grant et al., 1996; Kawabata, 2008). E2GMM (Chen & Ludtke, 2021) was among the first to apply GMMs to cryo-EM heterogeneous reconstruction. Like neural fields, GMMs can approximate any smooth density function and support differentiable optimization. More importantly, GMMs provide an explicit and interpretable representation that naturally links to atomic structures. Recent studies (Chen et al., 2023a;b; Li et al., 2024; Ducrocq et al., 2024; Schwab et al., 2024; Chen, 2025; Shekarforoush et al., 2025) have shown that GMMs can capture molecular flexibility by modeling atomic motion directly, making them highly suitable for heterogeneous reconstruction and downstream structural analysis.

However, existing GMM-based methods typically require initialization from an externally reconstructed consensus map or even an atomic model. Without such guidance, random initialization leads to unstable optimization and poor reconstruction quality. Our work addresses this limitation by introducing a GMM-based reconstruction pipeline that can be stably trained from scratch.

## 2.2 GAUSSIAN SPLATTING

3DGS (Kerbl et al., 2023) is a recent differentiable rendering technique developed for real-time novel view synthesis. It represents a 3D scene as a collection of anisotropic Gaussians and renders images via rasterization-based accumulation and alpha blending (Zwicker et al., 2002). While 3DGS achieves high visual fidelity in synthetic and real-world RGB datasets, as a volume rendering method, it is not a physically accurate model of natural image formation (Huang et al., 2024).

Although the original 3DGS formulation is not physically consistent with natural image formation, its volume rendering framework closely aligns with the cryo-EM imaging model, where each image arises from an orthographic line integral of electrostatic potential modulated by the contrast transfer function (CTF). Leveraging this alignment, we adapt splatting to cryo-EM by rederiving the projection of anisotropic Gaussians under cryo-EM physics, replacing heuristic alpha blending with physically accurate line integrals and incorporating CTF modulation.

## 3 METHOD

## 3.1 OVERVIEW

Our goal is to achieve physically accurate and computationally efficient cryo-EM reconstruction by leveraging a Gaussian Mixture Model (GMM). To this end, we propose cryoSplat, a differentiable framework that represents the 3D electrostatic potential of a specimen as a set of anisotropic Gaussians and directly simulates the cryo-EM image formation process in real space, faithfully adhering to the physics of transmission electron microscopy.

Building upon recent progress in differentiable volume rendering, particularly the Gaussian splatting framework by Kerbl et al. (2023), we adopt a tile-based rasterization strategy for scalable and effi-

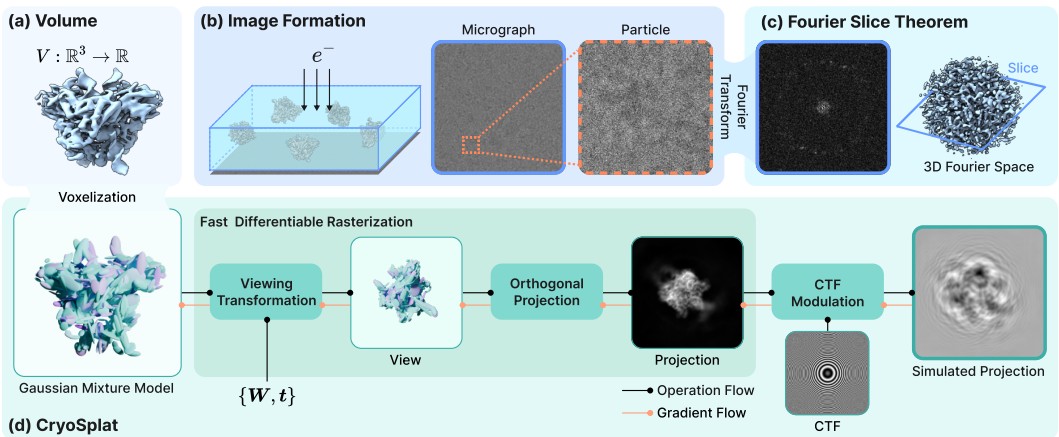

Figure 1: Cryo-EM reconstruction aims to recover a 3D volume (a) from a large set of 2D particle images (b). **(b)** Purified biomolecules with random orientations are embedded in a thin layer of vitreous ice. The electrostatic potential of the sample interacts with transmitted electrons, forming a micrograph that contains 2D projections of the molecules. Individual particle images are extracted from the micrograph; they are extremely noisy and modulated by highly oscillatory CTFs. **(c)** Fourier slice theorem: the 2D Fourier transform of a particle image corresponds to a central slice of the 3D Fourier transform of the volume. **(d)** Overview of cryoSplat. An anisotropic GMM representing the 3D volume is transformed to the projection direction, orthogonally projected onto a 2D image plane using a fast differentiable rasterizer, and modulated by the oscillatory CTF to simulate a physically accurate projection. The GMM parameters are optimized via gradients propagated from the discrepancy between the simulated and observed particle images. The resulting GMM can be voxelized to obtain the final 3D volume.

cient computation. However, the original 3DGS formulation is not directly applicable to cryo-EM due to several fundamental mismatches: **(i)** it employs perspective projection consistent with a pinhole camera model, in contrast to the orthographic projection in cryo-EM imaging; **(ii)** it is tailored for novel-view synthesis (i.e., photorealistic 2D appearance) rather than for physical 3D density reconstruction required in cryo-EM; and **(iii)** its image-centered coordinate system is incompatible with the FFT-aligned conventions assumed in cryo-EM reconstruction.

To address these issues, cryoSplat introduces several key adaptations: **(i)** we replace heuristic alpha blending with physically grounded line integrals to reflect the transmission nature of electron imaging; **(ii)** we fix the normalization between 3D-to-2D transformation and apply consistent learning rates across all parameters to ensure stable optimization; and **(iii)** we align the rasterization coordinate system with the FFT grid, allowing accurate gradient propagation through CTF modulation.

These modifications collectively enable cryoSplat to perform stable, end-to-end differentiable reconstruction from raw cryo-EM particle images, starting from random initialization without relying on externally provided volumes or atomic models.

## 3.2 IMAGE FORMATION

As shown in Fig. 1(b), electrons traverse a vitrified specimen, and the transmitted wavefronts undergo phase shifts due to the specimen's electrostatic potential (Singer & Sigworth, 2020). Under the weak phase approximation, the phase shifts are linearly related to the 3D potential (volume), and the image formed at the detector is a line integral (projection) of this potential along the beam direction, further convolved with $H : \mathbb{R}^2 \to \mathbb{R}$, the point spread function (PSF) of the imaging system.

In homogeneous reconstruction, it is assumed that all particle images $Y : \mathbb{R}^2 \to \mathbb{R}$ correspond to identical copies of a single 3D volume $V : \mathbb{R}^3 \to \mathbb{R}$, and that any conformational or compositional heterogeneity is negligible. Under this assumption, the image formation model can be expressed as:

$$Y(r_x, r_y) = H(r_x, r_y) * \int_{\mathbb{R}} V(\boldsymbol{W}^\top \boldsymbol{r} + \boldsymbol{t}) dr_z + \epsilon, \tag{1}$$

where $\boldsymbol{r} = [r_x, r_y, r_z]^\top$ are the 3D Cartesian coordinates in real space, $\boldsymbol{W} \in \mathrm{SO}(3)$ is the 3D pose of the particle, and $\boldsymbol{t} = [t_x, t_y, 0]^\top$ is the in-plane translation, accounting for imperfect centering during particle cropping. The noise term $\epsilon$ is modeled as independent, zero-mean Gaussian noise.

### 3.3 CRYOSPLAT

#### 3.3.1 ANISOTROPIC GMM

Anisotropic GMMs are developed to represent the volume, which can be written in the form

$$V(\boldsymbol{r}) = \sum_{i=1}^{N} A_i G_i(\boldsymbol{r}), \tag{2}$$

where $N$ denotes the Guassian count and $A_i$ is the amplitude of the $i$-th normalized Gaussian $G_i(\boldsymbol{r})$.

By substituting Eq. (1), we obtain the full forward process of cryoSplat. Specifically, we apply a viewing transformation to align the GMM to the target orientation, orthographically project each Gaussian along the $z$-axis to form a 2D image, and convolve the result with the PSF:

$$X(r_x, r_y) = H(r_x, r_y) * \sum_{i=1}^{N} A_i \int_{\mathbb{R}} G_i(\boldsymbol{W}^\top \boldsymbol{r} + \boldsymbol{t}) dr_z. \tag{3}$$

Since the integral is linear, each Gaussian contributes independently to the final image. We thus focus on a single Gaussian and omit the subscript $i$ in the following discussion. A normalized 3D Gaussian is defined as:

$$G(\boldsymbol{r}|\boldsymbol{\mu}, \boldsymbol{\Sigma}) = \frac{1}{(2\pi)^{\frac{3}{2}} |\boldsymbol{\Sigma}|^{\frac{1}{2}}} \exp\left( -\frac{1}{2}(\boldsymbol{r} - \boldsymbol{\mu})^\top \boldsymbol{\Sigma}^{-1}(\boldsymbol{r} - \boldsymbol{\mu}) \right), \tag{4}$$

where $\boldsymbol{\mu} \in \mathbb{R}^3$ and $\boldsymbol{\Sigma} \in \mathbb{R}^{3 \times 3}$ denote the mean (position) and the covariance matrix (shape), respectively. The determinant $|\boldsymbol{\Sigma}|$ ensures proper normalization. Following Kerbl et al. (2023), to guarantee the positive semidefinite property, we construct the covariance matrix as:

$$\boldsymbol{\Sigma} = \boldsymbol{R}\boldsymbol{S}\boldsymbol{S}^\top \boldsymbol{R}^\top, \tag{5}$$

where $\boldsymbol{S} = \mathrm{diag}(\boldsymbol{s})$ is a diagonal scaling matrix and $\boldsymbol{R} \in \mathrm{SO}(3)$ is a rotation matrix. In our implementation, we store the scaling vector $\boldsymbol{s} = [s_x, s_y, s_z]^\top$ and parameterize $\boldsymbol{R}$ using a quaternion $\boldsymbol{q} = [q_w, q_x, q_y, q_z]^\top$. To ensure positivity and stable gradients during optimization, we apply a softplus function to both the amplitude $A$ and the scaling vector $\boldsymbol{s}$. The quaternion $\boldsymbol{q}$ is normalized to ensure it represents a valid rotation. Altogether, each anisotropic Gaussian is parameterized by the 11-dimensional set $\{\mu_x, \mu_y, \mu_z, s_x, s_y, s_z, q_w, q_x, q_y, q_z, A\}$.

#### 3.3.2 VIEWING TRANSFORMATION

The viewing transformation is the first step in simulating image formation, aligning each Gaussian with a given projection direction. Since the parameters $\boldsymbol{\mu}$ and $\boldsymbol{\Sigma}$ describe Gaussians in world coordinates, we must transform them into the image-space coordinates before projection.

According to the derivation in Zwicker et al. (2002), applying an affine transformation to a Gaussian results in another Gaussian with appropriately transformed parameters. In our case, the transformation consists of a rotation $\boldsymbol{W} \in \mathrm{SO}(3)$ and a 2D in-plane translation $\boldsymbol{t} \in \mathbb{R}^3$, leading to:

$$\dot{G}(\boldsymbol{r}|\dot{\boldsymbol{\mu}}, \dot{\boldsymbol{\Sigma}}) = G(\boldsymbol{W}^\top \boldsymbol{r} + \boldsymbol{t}|\boldsymbol{\mu}, \boldsymbol{\Sigma}), \tag{6}$$

where the transformed mean and covariance are given by $\dot{\boldsymbol{\mu}} = \boldsymbol{W}(\boldsymbol{\mu} - \boldsymbol{t})$ and $\dot{\boldsymbol{\Sigma}} = \boldsymbol{W}\boldsymbol{\Sigma}\boldsymbol{W}^\top$.

#### 3.3.3 ORTHOGONAL PROJECTION

The orthogonal projection closely aligns with the physical principles of cryo-EM. Mathematically, it corresponds to a line integral of a 3D Gaussian along the $z$-axis, resulting in a 2D Gaussian, hereafter referred to as a splat, $\tilde{G}(\tilde{\boldsymbol{r}}|\tilde{\boldsymbol{\mu}}, \tilde{\boldsymbol{\Sigma}})$:

$$\tilde{G}(\tilde{\boldsymbol{r}}|\tilde{\boldsymbol{\mu}}, \tilde{\boldsymbol{\Sigma}}) = \int_{\mathbb{R}} \dot{G}(\boldsymbol{r}|\dot{\boldsymbol{\mu}}, \dot{\boldsymbol{\Sigma}}) dr_z. \tag{7}$$

This operation effectively integrates the 3D Gaussian along the projection axis, preserving its Gaussian form in 2D. The resulting closed-form expression is:

$$\tilde{G}(\tilde{\boldsymbol{r}}|\tilde{\boldsymbol{\mu}}, \tilde{\boldsymbol{\Sigma}}) = \frac{1}{2\pi|\tilde{\boldsymbol{\Sigma}}|^{\frac{1}{2}}} \exp\left(-\frac{1}{2}(\tilde{\boldsymbol{r}} - \tilde{\boldsymbol{\mu}})^\top \tilde{\boldsymbol{\Sigma}}^{-1}(\tilde{\boldsymbol{r}} - \tilde{\boldsymbol{\mu}})\right), \tag{8}$$

where $\tilde{\boldsymbol{r}} = [r_x, r_y]^\top$ denotes the 2D Cartesian coordinates in real space.

In prior works, such as 3DGS, the normalization term $1/(2\pi|\tilde{\boldsymbol{\Sigma}}|^{\frac{1}{2}})$ is often omitted, as their primary focus is on photorealistic novel view synthesis rather than the physical fidelity of the underlying 3D representation. However, in cryo-EM reconstruction, the ultimate goal is to recover the correct 3D volume. Omitting this view-dependent normalization introduces bias in amplitude and leads to incorrect reconstructions. Therefore, unlike 3DGS, we retain the normalization term to preserve the quantitative correctness of the model.

After projection, the final image is constructed by summing the weighted contributions of all splats and applying the PSF:

$$X(r_x, r_y) = H(r_x, r_y) * \sum_{i=1}^{N} A_i \tilde{G}_i(\tilde{\boldsymbol{r}}). \tag{9}$$

### 3.3.4 FAST DIFFERENTIABLE RASTERIZATION

We adopt the efficient tile-based rasterization framework from Kerbl et al. (2023), which enables scalable and differentiable processing of tens of thousands of Gaussians via per-tile accumulation. Unlike 3DGS, which uses alpha blending for photorealistic rendering, we modify the rasterization to directly sum contributions of splats, in accordance with the physical transmission model in cryo-EM.

For an image $\boldsymbol{X} \in \mathbb{R}^{D \times D}$, the original 3DGS implementation places the continuous coordinate center at $[(D-1)/2, (D-1)/2]^\top$, i.e., halfway between two discrete pixels. In contrast, FFT-based image formation assumes the origin is located at the integer grid point $[\lfloor D/2 \rfloor, \lfloor D/2 \rfloor]^\top$. To ensure compatibility with FFT-based forward and backward modeling, we shift the rasterization coordinates by half a pixel so that the image center aligns with the FFT grid. This alignment eliminates phase inconsistencies and enables accurate electron projection simulation, while preserving the computational efficiency of the 3DGS architecture. Let $\boldsymbol{X}, \boldsymbol{Y} \in \mathbb{R}^{D \times D}$ be the matrices representing the GMM-based projection $X$ and the observed image $Y$ after rasterization, respectively.

### 3.3.5 LOSS FUNCTION

Unlike previous GMM-based methods that rely on specially designed losses with complex regularization or constraints to ensure stable optimization, we adopt a much simpler formulation. Specifically, we directly apply the mean squared error (MSE) loss between the GMM-based projection $\boldsymbol{X}$ and the observed image $\boldsymbol{Y}$: $\mathcal{L} = \|\boldsymbol{X} - \boldsymbol{Y}\|_2^2$. Despite its simplicity, this loss formulation leads to stable and fast convergence in practice, without requiring additional regularization terms.

## 4 EXPERIMENT

### 4.1 EXPERIMENTAL SETTINGS

**Datasets.** We evaluate our method on four publicly available cryo-EM datasets from the Electron Microscopy Public Image Archive (EMPIAR) (Iudin et al., 2016): EMPIAR-10028 (*Pf*80S ribosome) (Wong et al., 2014), EMPIAR-10049 (RAG complex) (Ru et al., 2015), EMPIAR-10076 (*E. coli* LSU assembly) (Davis et al., 2016), and EMPIAR-10180 (pre-catalytic spliceosome) (Plaschka et al., 2017). These datasets span a range of structural complexity and image quality, from rigid assemblies with high contrast to highly heterogeneous macromolecular machines. For each dataset, we use the provided particle images, consensus pose estimates, and CTF parameters. All reconstructions are performed under the homogeneous assumption.

**Evaluation metrics.** To evaluate reconstruction quality on real datasets without ground truth volumes, we adopt the gold standard Fourier Shell Correlation (FSC) (Van Heel & Schatz, 2005), following established protocols. Each dataset is randomly split into two halves, and the reconstruction

algorithm is applied independently to each subset. Let the resulting volumes be $\widehat{V}_a(\boldsymbol{k})$ and $\widehat{V}_b(\boldsymbol{k})$, representing their Fourier transforms. The FSC is computed as a function of frequency $k$ using the following formula:

$$\text{FSC}(k) = \frac{\sum\limits_{\|\boldsymbol{k}\|_2 = k} \widehat{V}_a(\boldsymbol{k}) \cdot \widehat{V}_b(\boldsymbol{k})^*}{\sqrt{\left( \sum\limits_{\|\boldsymbol{k}\|_2 = k} |\widehat{V}_a(\boldsymbol{k})|^2 \right) \left( \sum\limits_{\|\boldsymbol{k}\|_2 = k} |\widehat{V}_b(\boldsymbol{k})|^2 \right)}}. \tag{10}$$

This metric quantifies the correlation between two independently reconstructed volumes within concentric shells in Fourier space. The spatial resolution is defined as the frequency where the FSC curve drops below the $0.143$ threshold, indicating the limit of reproducible structural detail.

**Implementation details.** For all experiments, particle images from EMPIAR-10028, 10076, and 10180 are downsampled to $256 \times 256$, while EMPIAR-10049 is used at its original $192 \times 192$ resolution. Published particle translations are applied to the observed images via phase shifting in Fourier space prior to reconstruction, rather than through the GMM viewing transform. We do not apply any windowing to the observed particle images during preprocessing. The 3D volume is defined over the domain $[-E, E]^3$, and each 2D projection is assumed to lie within $[-E, E]^2$ in the image plane, where $E = 0.5$ defines the spatial extent. The values used in initialization are fixed but grounded in straightforward statistical intuition. We observe that most particles are concentrated within a spherical region of radius $E/2$. To reflect this prior and accelerate convergence, we initialize the Gaussian means within this region. More specifically, based on the three-sigma rule for Gaussian distributions $\mathcal{N}(\mu, \sigma^2)$, where $99.7\%$ of samples fall within $[\mu - 3\sigma, \mu + 3\sigma]$, we obtain $\sigma = E/6$ from $3\sigma = E/2$. To slightly tighten the spread, we apply a scaling factor and use $\sigma = 0.9 \cdot E/6 = 0.075$ to initialize the means. The initial scale of each Gaussian is set to $0.1 \times 0.075 = 0.0075$, encouraging localized support. Finally, to maintain consistent overall energy across varying numbers of Gaussians, we initialize the amplitude as $A = 1/(2N)$, where $N$ is the total number of Gaussians. All parameters are trainable. In the original 3DGS (Kerbl et al., 2023), different learning rates are assigned to different types of Gaussian parameters (means, scales, rotations, opacities). While this works well in novel view synthesis, it introduces instability in cryo-EM reconstruction. Let the full parameter vector be $\boldsymbol{\theta} = [\mu_x, \mu_y, \mu_z, s_x, s_y, s_z, q_w, q_x, q_y, q_z, A]^\top$. In gradient descent optimization, the direction of parameter updates is determined by the gradient $\nabla_{\boldsymbol{\theta}} \mathcal{L}$. Unequal learning rates distort this direction by scaling different components unequally, which can lead to divergence. We observe that such practice causes Gaussians to spread uncontrollably in early iterations and finally diverge. To avoid this, we adopt a single unified learning rate across all parameter types, preserving the intended descent direction and ensuring stable convergence. Accordingly, we use Adam (Kingma & Ba, 2014) with batch size 1, learning rate 0.001, and exponential decay ($\gamma = 0.1$) at each epoch. All GMMs are trained for 5 epochs. For the voxel-based baseline, we run cryoSPARC's "Homogeneous Reconstruction Only" job using the same poses and CTF parameters. For neural representation learning, we follow cryoDRGN's default configuration: three 1,024-node layers, trained for 50 epochs. All experiments are run on a single NVIDIA GeForce RTX 3090.

## 4.2 EVALUATION ON REAL DATASETS

We evaluate the performance of different volume representations on real cryo-EM datasets under a homogeneous reconstruction setting. To ensure a fair comparison focused solely on the choice of volume representation, all methods reconstruct consensus maps using the same set of particle poses published by Zhong et al. (2021a), without performing pose estimation. Our evaluation focuses on two aspects: (i) the ability to reconstruct high-quality consensus maps, and (ii) robustness to noise and imperfect pose assignments. Since related methods (Zhong et al., 2021a;b; Levy et al., 2022a;b; 2025) adopt cryoDRGN's neural field implementation and differ mainly in pose estimation, we focus our comparison on cryoDRGN. Accordingly, we evaluate three representative approaches: voxel-based backprojection cryoSPARC (Punjani et al., 2017), the neural representation method cryoDRGN (Zhong et al., 2021a), and our proposed GMM-based method cryoSplat. Visualizations of the reconstructed volumes are shown in Fig. 2, and spatial resolution is quantified using gold-standard FSC curves. Each volume of cryoSplat is represented using 30,000 Gaussians.

The *Pf* 80S ribosome (EMPIAR-10028) is relatively easy to reconstruct due to its high-contrast images and structurally stable particles. All methods achieve high-resolution results (3.80 Å) and

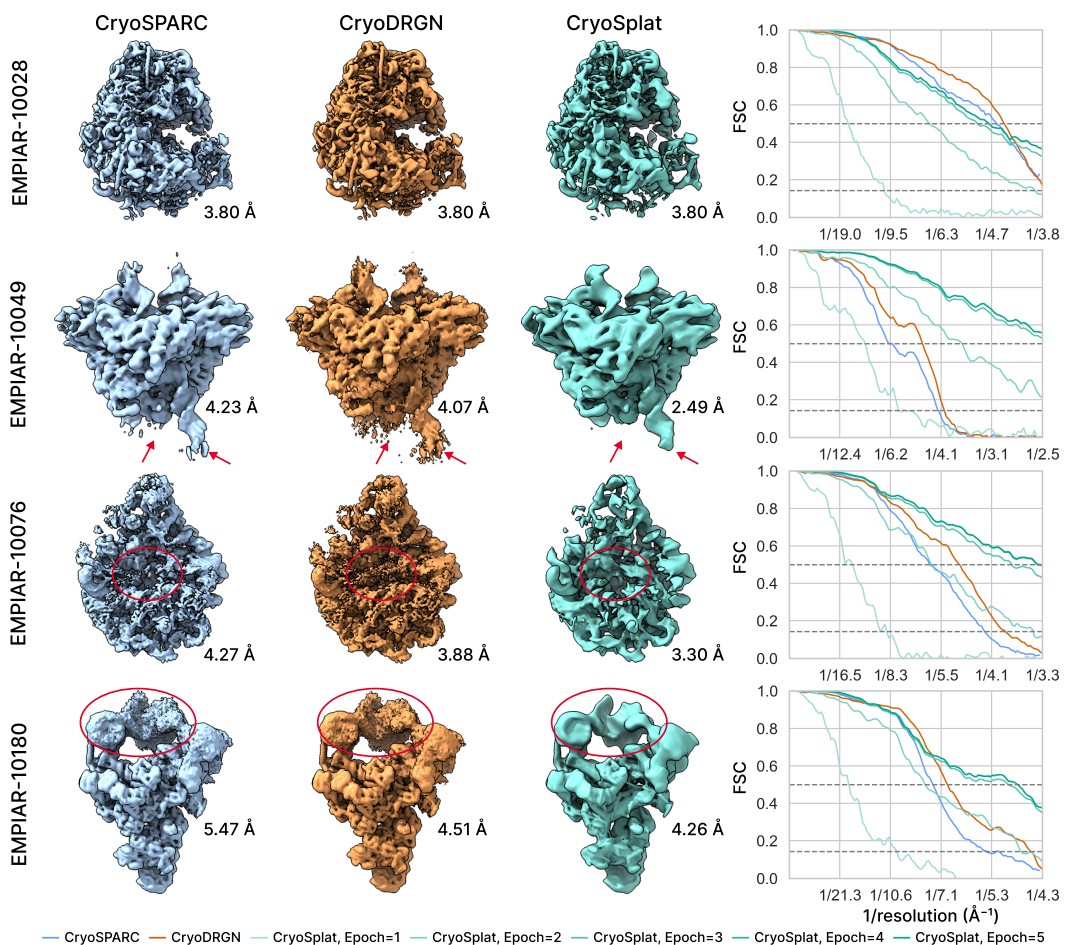

Figure 2: Qualitative and quantitative comparison of voxel-based, neural, and GMM-based representations. (**Left**) Final 3D reconstructions on four real datasets visualized with ChimeraX (Pettersen et al., 2021). (**Right**) FSC curves are plotted for quantitative evaluation. Gray dashed lines indicate the standard resolution thresholds of 0.5 and 0.143, reported in Angstroms (Å). CryoSplat consistently achieves higher resolution across all datasets.

strong FSC agreement across the spectrum. cryoDRGN yields slightly higher FSC values at intermediate frequencies, while cryoSplat outperforms all baselines at high spatial frequencies, demonstrating its ability to recover fine structural details.

The RAG complex (EMPIAR-10049) poses greater challenges due to symmetry-induced pose degeneracy and flexible regions such as the DNA elements and the nonamer binding domain (NBD), indicated by arrows. CryoSplat outperforms the baselines with a higher FSC, achieving a resolution of 2.49 Å. Unlike the baselines, cryoSplat reconstructs the DNA elements and the NBD with minimal density fragments. Its FSC curve remains consistently above those of other methods across all spatial frequencies, highlighting its robustness to pose degeneracy and structural variability.

This LSU assembly dataset (EMPIAR-10076) contains substantial compositional and conformational heterogeneity, making consensus reconstruction particularly challenging. Both FSC analysis and visualization show that cryoSplat is more resilient under such conditions, achieving a resolution of 3.30 Å with fewer fragments than voxel-based or neural methods, as indicated by the red circle.

The spliceosome dataset (EMPIAR-10180) features large-scale motions of the SF3b indicated by the red circle, making consensus reconstruction particularly challenging. The reconstructions from cryoSPARC and cryoDRGN show pronounced high-frequency spurious spikes in this region, while

cryoSplat is more robust to such motions and achieves a resolution of $4.26$ Å. FSC analysis further confirms that cryoSplat significantly outperforms the baselines across the frequency range.

CryoSplat consistently converges within 5 epochs, with FSC curves from the 4th and 5th epochs tightly overlapping, indicating stable optimization and improved generalization.

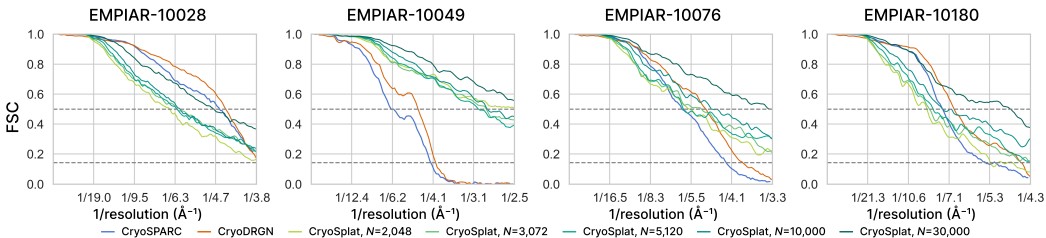

Figure 3: Reconstruction performance with varying numbers of Gaussians. Increasing the number improves accuracy and robustness.

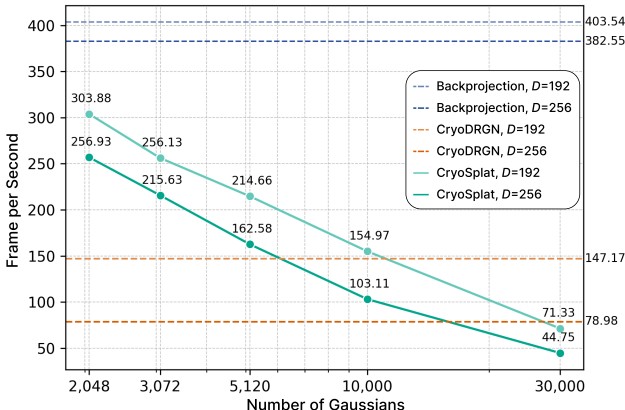

Figure 4: Runtime efficiency across reconstruction methods at different resolutions ($D = 192$ and $D = 256$). Frame rates (FPS) are measured under increasing numbers of Gaussians (log-scaled).

### 4.3 ABLATION STUDIES

This section reports ablation studies of our approach. More results can be found in Appendix D.

**Number of Gaussians.** Fig. 3 shows the FSC curves for cryoSplat with varying numbers of Gaussians. In general, increasing the number of Gaussians leads to improved FSC, as a denser GMM provides greater representational capacity. While cryoSplat performs well on most datasets, its relative performance varies due to differences in structural complexity and dataset-specific challenges. On EMPIAR-10028, cryoSplat reaches a resolution of 3.8 Å under all settings. While configurations with fewer than $10,000$ Gaussians exhibit lower FSC values than cryoDRGN and cryoSPARC across most frequencies, the curves intersect at the highest frequency, indicating comparable final resolution. For EMPIAR-10049, all cryoSplat settings significantly outperform both cryoSPARC ($4.23$ Å) and cryoDRGN ($4.07$ Å), achieving a resolution of $2.49$ Å. Moreover, the FSC curves of all cryoSplat variants remain consistently above those of the two baselines across the entire frequency range. For EMPIAR-10076, the 30,000-Gaussian model clearly outperforms other settings; even with fewer Gaussians, cryoSplat still surpasses the baselines, reaching 3.3 Å. For EMPIAR-10180, the models with $10,000$ and $30,000$ Gaussians achieve the best FSC, reaching $4.3$ Å, while sparser GMMs remain competitive at high spatial frequencies. Overall, we observe that using $10,000$ Gaussians is sufficient to provide a robust improvement in FSC-derived resolution metrics over baseline methods across most datasets. Associated qualitative comparisons are provided in Appendix D.1.

Table 1: GPU memory usage across reconstruction methods at resolutions ($D = 192$, $D = 256$).

| Methods | # Params | Settings | Batch Size | GPU Mem. (MiB) $D = 192$ | $D = 256$ |
|---|---|---|---|---|---|
| Backprojection | $(D + 1)^3$ | — | 1 | 508 | 396 |
| CryoDRGN (Zhong et al., 2021a) | $(6 \cdot \lfloor D/2 \rfloor + L + 3) \cdot C$ $+L \cdot C^2 + 2$ | $C = 1{,}024$ $L = 3$ | 1 8 | 680 2,560 | 1,008 4,906 |
| CryoSplat (Ours) | $11 \cdot N$ | $N = 2{,}048$ $N = 3{,}072$ $N = 5{,}120$ $N = 10{,}000$ $N = 30{,}000$ | 1 | 344 344 346 348 376 | 346 348 348 350 378 |

**Runtime efficiency.** We compare the runtime efficiency of cryoSplat with other representation baselines, as shown in Fig. 4. Backprojection is the fastest, as it generates projections by directly indexing and interpolating from a dense voxel grid, but its cubic scaling makes it unsuitable for modern non-linear heterogeneous analysis. For such tasks, neural representations and GMMs offer greater flexibility. Under commonly used settings in heterogeneous reconstruction (e.g., 2,048–3,072 Gaussians (Chen & Ludtke, 2021; Chen et al., 2023a;b)), cryoSplat achieves 2–3× higher FPS than cryoDRGN. Moreover, cryoSplat typically converges within 5 epochs, compared to 50 epochs required by cryoDRGN, providing an overall speedup up to 30×. As discussed above, using 10,000 Gaussians allows cryoSplat to consistently outperform baseline methods in FSC across most datasets, while still maintaining a higher FPS than cryoDRGN. Even with an extremely large number of Gaussians (e.g., 30,000), cryoSplat provides reasonable runtime performance for orthogonal projection operations. Overall, as shown in Fig. 4, cryoSplat demonstrates sub-linear time complexity with respect to the number of Gaussians, offering a favorable trade-off between accuracy and efficiency.

**Memory usage.** Tab. 1 compares GPU memory usage across different reconstruction methods. CryoDRGN (Zhong et al., 2021a) exhibits the highest memory footprint, exceeding 2.5 GiB at $D = 192$ and approaching 5 GiB at $D = 256$, primarily due to its deep neural decoder and a larger batch size of 8. Interestingly, backprojection consumes more memory at $D = 192$ than at $D = 256$, which may be attributed to implementation-specific factors such as padding overhead or kernel-level optimizations favoring power-of-two dimensions. This anomaly appears method-specific and does not reflect a general trend. In contrast, cryoSplat demonstrates consistently low memory usage across all configurations. Even with as many as 30,000 Gaussians, cryoSplat maintains a footprint below 380 MiB, with negligible variation across resolutions. This efficiency underscores the scalability and suitability of cryoSplat for large-scale or memory-constrained cryo-EM reconstruction scenarios.

## 5 CONCLUSION

We present cryoSplat, a novel GMM-based framework that integrates Gaussian splatting with the physics of cryo-EM image formation. CryoSplat enables stable and efficient homogeneous reconstruction directly from raw cryo-EM particle images, starting from random initialization without relying on consensus volumes. By operating on an anisotropic Gaussian representation under a principled formulation, cryoSplat mitigates instability issues commonly encountered in GMM-based optimization and provides a fully differentiable reconstruction framework. Experimental results on real datasets demonstrate the effectiveness, robustness, and faster convergence of cryoSplat compared to representative baselines.

**Limitation and future work**. The current formulation of cryoSplat assumes known particle poses and therefore does not yet constitute an *ab initio* reconstruction method. While this limits its applicability in fully unsupervised settings, cryoSplat establishes a principled and stable foundation for integrating GMMs into cryo-EM reconstruction under realistic imaging physics. In particular, the proposed framework improves optimization stability and initialization robustness for Gaussian-based models. Promising future directions include jointly optimizing poses and Gaussian parameters, extending the framework to heterogeneous reconstruction, and integrating cryoSplat into *ab initio* end-to-end cryo-EM pipelines.

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

APPENDIX

# A    DETAILS OF METHOD

## A.1    REAL SPACE RECONSTRUCTION

According to the Fourier slice theorem (Bracewell, 1956), illustrated in Fig. 1(c), the 2D Fourier transform of a projection corresponds to a central slice of the 3D Fourier transform of the volume, orthogonal to the projection direction and passing through the origin. Based on this property, an alternative and widely adopted formulation models reconstruction directly in the Fourier domain, where the image formation model becomes:

$$\widehat{Y}(k_x, k_y) = \widehat{H}(k_x, k_y) \cdot \widehat{V}(\boldsymbol{W}^\top \boldsymbol{k}) \cdot e^{-2\pi i \boldsymbol{k}^\top \boldsymbol{t}} + \widehat{\epsilon}, \tag{11}$$

where $\boldsymbol{k} = [k_x, k_y, 0]^\top$ denotes the Cartesian coordinates in Fourier space, and the 2D spectrum $\widehat{Y}$, the CTF $\widehat{H}$ and the 3D spectrum $\widehat{V}$ denote the Fourier transform of $Y$, $H$ and $V$, respectively. The noise term $\widehat{\epsilon}$ is similarly modeled as independent, zero-mean Gaussian noise in the Fourier domain.

In this work, departing from most existing approaches that adopt Eq. (11), we instead build our pipeline on Eq. (1), performing homogeneous reconstruction directly in real space.

A key reason we choose to operate in real space is that it allows us to fully exploit the fast rasterization strategy from 3DGS. In high-resolution reconstructions, individual Gaussians in real space have small spatial scales and affect only a few nearby tiles, as shown in Fig. 5(a). This locality means that each GPU thread is responsible for a single pixel and only needs to process a small subset of all Gaussians. In contrast, Gaussians in Fourier space become broad as resolution increases, leading to near-global support. As a result, each pixel in the frequency domain must aggregate contributions from nearly all Gaussians, making fast rendering impractical.

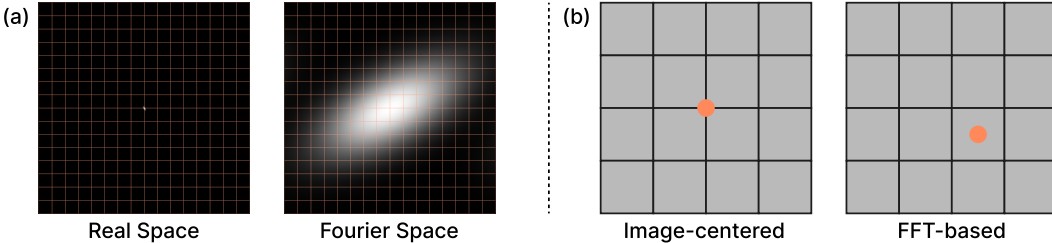

(a) Real Space    Fourier Space    (b) Image-centered    FFT-based

Figure 5: Rasterization details. **(a)** A Gaussian with small spatial scales in real space during high-resolution reconstruction overlaps at most four tiles, while in Fourier space it exhibits nearly global support. Tile boundaries are indicated by lines. **(b)** For a $4 \times 4$ image, the origin of the continuous coordinate system during rasterization is defined differently: FFT-based coordinates place it at $[2, 2]^\top$, whereas image-centered coordinates place it at $[1.5, 1.5]^\top$. The origin is marked by a dot. Image-centered coordinates induce a phase error of $-\pi \boldsymbol{k}/D$, up to $\pm\pi/2$ at the Nyquist frequency, degrading reconstruction at high frequency. The effect diminishes with larger $D$ but never vanishes.

## A.2    COMPUTATIONAL DETAILS

Section 3.3.3 describes how a 3D Gaussian is projected along the $z$-axis to form a splat. As discussed in Zwicker et al. (2002), the splat can be computed analytically by removing the $z$-axis components from the mean and covariance:

$$\begin{cases} \dot{\boldsymbol{\mu}} = [\dot{\mu}_x, \dot{\mu}_y, \dot{\mu}_z]^\top \Rightarrow [\dot{\mu}_x, \dot{\mu}_y]^\top = \tilde{\boldsymbol{\mu}} \\ \dot{\boldsymbol{\Sigma}} = \begin{bmatrix} \dot{\sigma}_{xx} & \dot{\sigma}_{xy} & \dot{\sigma}_{xz} \\ \dot{\sigma}_{xy} & \dot{\sigma}_{yy} & \dot{\sigma}_{yz} \\ \dot{\sigma}_{xz} & \dot{\sigma}_{yz} & \dot{\sigma}_{zz} \end{bmatrix} \Rightarrow \begin{bmatrix} \dot{\sigma}_{xx} & \dot{\sigma}_{xy} \\ \dot{\sigma}_{xy} & \dot{\sigma}_{yy} \end{bmatrix} = \tilde{\boldsymbol{\Sigma}} \end{cases} \tag{12}$$

thereby enabling an efficient computation of Eq. (7).

As discussed in Sec. 3.3.4 and shown in Fig. 5(b), when a continuous image $X : \mathbb{R}^2 \to \mathbb{R}$ is rasterized onto pixels $\boldsymbol{X} \in \mathbb{R}^{D \times D}$, the origin of the continuous coordinate system should be aligned with $[\lfloor D/2 \rfloor, \lfloor D/2 \rfloor]^\top$ to match the FFT-based coordinate convention used in cryo-EM. Formally,

$$X_{i,j} = X\left((j - \lfloor \frac{D}{2} \rfloor)\frac{2E}{D}, -(i - \lfloor \frac{D}{2} \rfloor)\frac{2E}{D}\right), \tag{13}$$

where $X_{i,j}$ denotes $(i,j)$-th entry of matrix $\boldsymbol{X}$. Note that the row and column indices correspond to the $y$- and $x$-axes, respectively, with the $y$-axis flipped during this rasterization.

In practice, since the PSF corresponds to a large convolution kernel, we apply the contrast transfer function (CTF) in the Fourier domain after rasterization for efficiency:

$$X_{i,j} = \mathcal{F}^{-1}\left(\widehat{\boldsymbol{H}} \odot \mathcal{F}\left(\sum_{i=1}^{N} A_i \tilde{G}_i \left((j - \lfloor \frac{D}{2} \rfloor)\frac{2E}{D}, -(i - \lfloor \frac{D}{2} \rfloor)\frac{2E}{D}\right)\right)\right), \tag{14}$$

where $\mathcal{F}(\cdot)$ and $\mathcal{F}^{-1}(\cdot)$ denote the Fourier and inverse Fourier transform operators, respectively. $\widehat{\boldsymbol{H}}$ is the rasterized CTF $\widehat{H}$ and $\odot$ denotes element-wise (Hadamard) product.

Before deriving the gradients, we first define

$$Q(r_x, r_y) = \sum_{i=1}^{N} A_i \tilde{G}_i(\tilde{\boldsymbol{r}}), \tag{15}$$

which is the pre-rasterization continuous image. For clarity, we omit the Gaussian index $i$ in the following derivations, as the gradients are computed independently for each Gaussian. We denote by $\mathbb{P}$ the set of 2D coordinates corresponding to the centers of rasterized pixels. When $\tilde{\boldsymbol{r}} \in \mathbb{P}$, the coordinate $\tilde{\boldsymbol{r}} = [r_x, r_y]^\top$ refers to a discrete sampling location in the image plane. The gradients used in the backward pass can be summarized as

$$\begin{cases} \dfrac{\partial \mathcal{L}}{\partial A} = \sum_{\tilde{\boldsymbol{r}} \in \mathbb{P}} \dfrac{\partial \mathcal{L}}{\partial Q(\tilde{\boldsymbol{r}})} \dfrac{\partial Q(\tilde{\boldsymbol{r}})}{\partial A} \\[2ex] \nabla_{\boldsymbol{\mu}} \mathcal{L} = \sum_{\tilde{\boldsymbol{r}} \in \mathbb{P}} \dfrac{\partial \mathcal{L}}{\partial Q(\tilde{\boldsymbol{r}})} \nabla_{\boldsymbol{\mu}} Q(\tilde{\boldsymbol{r}}) \\[2ex] \nabla_{\boldsymbol{s}} \mathcal{L} = \sum_{\tilde{\boldsymbol{r}} \in \mathbb{P}} \dfrac{\partial \mathcal{L}}{\partial Q(\tilde{\boldsymbol{r}})} \nabla_{\boldsymbol{\Sigma}} Q(\tilde{\boldsymbol{r}}) \circ \dfrac{\partial \boldsymbol{\Sigma}}{\partial \boldsymbol{s}} \\[2ex] \nabla_{\boldsymbol{q}} \mathcal{L} = \sum_{\tilde{\boldsymbol{r}} \in \mathbb{P}} \dfrac{\partial \mathcal{L}}{\partial Q(\tilde{\boldsymbol{r}})} \nabla_{\boldsymbol{\Sigma}} Q(\tilde{\boldsymbol{r}}) \circ \dfrac{\partial \boldsymbol{\Sigma}}{\partial \boldsymbol{q}} \end{cases} \tag{16}$$

where $\circ$ denotes the composition of Jacobian operators (chain rule). The derivation of gradients with respect to the amplitude $A$ and mean $\boldsymbol{\mu}$ is trivial, which can be given directly by

$$\frac{\partial Q}{\partial A} = \tilde{G}(\tilde{\boldsymbol{r}}), \tag{17}$$

and

$$\begin{cases} \nabla_{\tilde{\boldsymbol{\mu}}} Q = A \tilde{G}(\tilde{\boldsymbol{r}}) \tilde{\boldsymbol{\Sigma}}^{-1} (\tilde{\boldsymbol{r}} - \tilde{\boldsymbol{\mu}}) \\ \nabla_{\boldsymbol{\mu}} Q = W [\nabla_{\tilde{\boldsymbol{\mu}}} Q^\top \ 0]^\top \end{cases} \tag{18}$$

where $[\nabla_{\tilde{\boldsymbol{\mu}}} Q^\top \ 0]^\top$ embeds the 2D gradient into 3D space by padding the $z$-component with zero.

For completeness, we provide the derivation of the covariance gradients $\nabla_{\boldsymbol{\Sigma}} Q$, noting that our formulation retains the normalization term, which is omitted in 3DGS (Kerbl et al., 2023). Remember

$$\begin{aligned} \tilde{G}(\tilde{\boldsymbol{r}} | \tilde{\boldsymbol{\mu}}, \tilde{\boldsymbol{\Sigma}}) &= \frac{1}{2\pi |\tilde{\boldsymbol{\Sigma}}|^{\frac{1}{2}}} \exp\left(-\frac{1}{2}(\tilde{\boldsymbol{r}} - \tilde{\boldsymbol{\mu}})^\top \tilde{\boldsymbol{\Sigma}}^{-1} (\tilde{\boldsymbol{r}} - \tilde{\boldsymbol{\mu}})\right) \\ &= \frac{|\tilde{\boldsymbol{\Sigma}}^{-1}|^{\frac{1}{2}}}{2\pi} \exp\left(-\frac{1}{2}(\tilde{\boldsymbol{r}} - \tilde{\boldsymbol{\mu}})^\top \tilde{\boldsymbol{\Sigma}}^{-1} (\tilde{\boldsymbol{r}} - \tilde{\boldsymbol{\mu}})\right). \end{aligned} \tag{19}$$

We can first compute

$$
\begin{aligned}
\nabla_{\tilde{\boldsymbol{\Sigma}}^{-1}} Q &= A \exp(-\frac{1}{2}(\tilde{\boldsymbol{r}} - \tilde{\boldsymbol{\mu}})^\top \tilde{\boldsymbol{\Sigma}}^{-1}(\tilde{\boldsymbol{r}} - \tilde{\boldsymbol{\mu}}))\frac{1}{4\pi}|\tilde{\boldsymbol{\Sigma}}^{-1}|^{-\frac{1}{2}}|\tilde{\boldsymbol{\Sigma}}^{-1}|\tilde{\boldsymbol{\Sigma}}^\top \\
&\quad + A\frac{|\tilde{\boldsymbol{\Sigma}}^{-1}|^{\frac{1}{2}}}{2\pi}\exp(-\frac{1}{2}(\tilde{\boldsymbol{r}} - \tilde{\boldsymbol{\mu}})^\top \tilde{\boldsymbol{\Sigma}}^{-1}(\tilde{\boldsymbol{r}} - \tilde{\boldsymbol{\mu}}))(-\frac{1}{2}(\tilde{\boldsymbol{r}} - \tilde{\boldsymbol{\mu}})(\tilde{\boldsymbol{r}} - \tilde{\boldsymbol{\mu}})^\top) \\
&= \tfrac{1}{2}A\tilde{G}(\tilde{\boldsymbol{r}})(\tilde{\boldsymbol{\Sigma}} - (\tilde{\boldsymbol{r}} - \tilde{\boldsymbol{\mu}})(\tilde{\boldsymbol{r}} - \tilde{\boldsymbol{\mu}})^\top),
\end{aligned}
\tag{20}
$$

and then $\nabla_{\tilde{\boldsymbol{\Sigma}}} Q = -\tilde{\boldsymbol{\Sigma}}^{-\top}\nabla_{\tilde{\boldsymbol{\Sigma}}^{-1}} Q \tilde{\boldsymbol{\Sigma}}^{-\top}$. Finally,

$$
\nabla_{\dot{\boldsymbol{\Sigma}}} Q = \begin{bmatrix} \nabla_{\tilde{\boldsymbol{\Sigma}}} Q & \mathbf{0} \\ \mathbf{0}^\top & 0 \end{bmatrix}.
\tag{21}
$$

The subsequent derivations of $\nabla_{\boldsymbol{s}}\mathcal{L}$ and $\nabla_{\boldsymbol{q}}\mathcal{L}$ follow exactly the formulation in Kerbl et al. (2023).

## B  DATASET DETAILS

We provide detailed statistics and characteristics of the cryo-EM datasets used in our experiments:

- EMPIAR-10028 (*Plasmodium falciparum* 80S (*Pf*80S) ribosome) (Wong et al., 2014): 105,247 particle images of size $360 \times 360$ pixels at a sampling rate of 1.34 Å/pixel. This is a widely used benchmark with high-contrast images and a static structure.

- EMPIAR-10049 (RAG1-RAG2 complex) (Ru et al., 2015): 108,544 particles of size $192 \times 192$ pixels at 1.23 Å/pixel. This dataset is considered more challenging due to its lower contrast and flexibility in some regions.

- EMPIAR-10076 (*E. coli* large ribosomal subunit undergoing (LSU) assembly) (Davis et al., 2016): 131,899 particles of size $320 \times 320$ pixels at 1.31 Å/pixel. This dataset contains substantial conformational and compositional heterogeneity, which poses a challenge to homogeneous modeling.

- EMPIAR-10180 (Pre-catalytic spliceosome) (Plaschka et al., 2017): 327,490 particles of size $320 \times 320$ pixels at 1.69 Å/pixel. It samples a continuum of conformations, particularly involving large-scale motions of the SF3b subcomplex.

- Synthetic 80S ribosome: We construct a synthetic dataset of the 80S ribosome with 100,000 particles using Relion (Scheres, 2016a), following the protocol of Levy et al. (2022a). The electron scattering potential is derived in ChimeraX (Pettersen et al., 2021) at a resolution of 6.0 Å/pixel, based on two atomic models: the small subunit (PDB 3J7A) and the large subunit (PDB 3J79) (Wong et al., 2014). Each particle image is $128 \times 128$ pixels with a pixel size of 3.77 Å/pixel. Orientations are uniformly sampled over $SO(3)$, and all images are centered without translations. Defocus values for the CTF are randomly drawn from log-normal distributions following Levy et al. (2022a), and zero-mean white Gaussian noise with varying signal-to-noise ratios (SNRs) is added.

## C  MORE IMPLEMENTATION DETAILS

### C.1  OPTIMIZATION ALGORITHM

Our optimization algorithm is summarized in Algorithm 1. Unlike Kerbl et al. (2023), which uses gradient magnitude as the criterion for splitting and cloning Gaussians, we observe that gradients are not a reliable indicator for densification in cryo-EM reconstruction. Furthermore, elaborate densification schemes are generally unnecessary, as our method seldom suffers from significant local minima owing to its close consistency with cryo-EM imaging physics. Nevertheless, we retain a simple densification option to balance efficiency and resolution: fewer Gaussians enable faster training, whereas more Gaussians yield higher-resolution reconstructions, as demonstrated in Fig. 4.

---

**Algorithm 1** Optimization and Densification
$N$: number of Gaussians
$D$: side length of the observed particle images

---

$\quad \Theta \leftarrow$ InitAttributes$(N)$ $\qquad\qquad$ ▷ Positions, Scales, Quaternions, Amplitudes
$\quad i \leftarrow 0$ $\qquad\qquad$ ▷ Epoch Count
$\quad$**while** not converged **do**
$\quad\quad$**for** $(Y, W, t, \widehat{H})$ **in** Dataloader() **do** $\qquad$ ▷ Observed Image, Rotation, Translation, CTF
$\quad\quad\quad Y \leftarrow$ FourierShift$(Y, t)$ $\qquad\qquad$ ▷ Center Alignment
$\quad\quad\quad Q \leftarrow$ Rasterize$(\Theta, W, D)$ $\qquad\qquad$ ▷ Algorithm 2
$\quad\quad\quad X \leftarrow$ ApplyCTF$(Q, \widehat{H})$ $\qquad\qquad$ ▷ Apply CTF
$\quad\quad\quad \mathcal{L} \leftarrow$ Loss$(X, Y)$ $\qquad\qquad$ ▷ Loss
$\quad\quad\quad \Theta \leftarrow$ Adam$(\nabla\mathcal{L})$ $\qquad\qquad$ ▷ Backprop and Step
$\quad\quad$**end for**
$\quad\quad$**if** IsDoubleGaussians$(i)$ **then** $\qquad\qquad$ ▷ (Optional) Densification
$\quad\quad\quad$**for all** Gaussian$(\mu, s, q, A)$ **in** $\Theta$ **do**
$\quad\quad\quad\quad$SplitGaussian$(\mu, s, q, A)$
$\quad\quad\quad$**end for**
$\quad\quad$**end if**
$\quad\quad i \leftarrow i + 1$
$\quad$**end while**

---

**Algorithm 2** CUDA-accelerated Rasterization
$\Theta$: Gaussian parameters
$W$: viewing transformation matrix
$D$: side length of the observed particle images

---

$\quad$**function** Rasterize$(\Theta, W, D)$
$\quad\quad \mu, \Sigma, A \leftarrow$ BuildGaussians$(\Theta)$
$\quad\quad \dot{\mu}, \dot{\Sigma} \leftarrow$ ViewingTransform$(\mu, \Sigma, W)$ $\qquad$ ▷ Viewing Transformation
$\quad\quad \tilde{\mu}, \tilde{\Sigma} \leftarrow$ Projection$(\dot{\mu}, \dot{\Sigma})$ $\qquad$ ▷ Orthogonal Projection
$\quad\quad T \leftarrow$ CreateTiles$(D)$ $\qquad$ ▷ Tile Count
$\quad\quad L, K \leftarrow$ DuplicateWithKeys$(\tilde{\mu}, T)$ $\qquad$ ▷ List of Indices and Keys
$\quad\quad$SortByKeys$(K, L)$
$\quad\quad R \leftarrow$ IdentifyTileRanges$(T, K)$
$\quad\quad Q \leftarrow 0$ $\qquad$ ▷ Init Canvas
$\quad\quad$**for all** Tile $t$ **in** $Q$ **do**
$\quad\quad\quad$**for all** Pixel $p$ **in** $t$ **do**
$\quad\quad\quad\quad r \leftarrow$ GetTileRange$(R, t)$
$\quad\quad\quad\quad Q(p) \leftarrow$ SumSplats$(p, L, r, K, \tilde{\mu}, \tilde{\Sigma}, A)$
$\quad\quad\quad$**end for**
$\quad\quad$**end for**
$\quad\quad$**return** $Q$
$\quad$**end function**

---

## C.2 DETAILS OF THE RASTERIZER

The details of the rasterizer are summarized in Algorithm 2. We follow the tile-based rasterization framework of Kerbl et al. (2023), where the output image is divided into $16 \times 16$ pixel tiles, and each splat is instantiated in every tile it overlaps. The splat instances are then assigned keys for sorting, after which each tile can be processed efficiently by locating the corresponding ranges in the sorted list. Since pixels are computed in parallel, the runtime is primarily determined by the maximum number of Gaussians within any tile. For more details, we refer the reader to Kerbl et al. (2023).

## C.3 DETAILS OF REPORTED METRICS

When computing FSC curves for baselines, a spherical mask is applied to suppress background noise; cryoSplat uses the unmasked FSC because its Gaussian representation naturally suppresses

noise outside the signal region. For runtime and memory comparisons, we use the backprojection implementation by Zhong et al. (2021a) instead of cryoSPARC, whose packaged environment introduces additional subprocesses and overhead that hinder fair measurement.

# D    ADDITIONAL EXPERIMENTS

## D.1    NUMBER OF GAUSSIANS

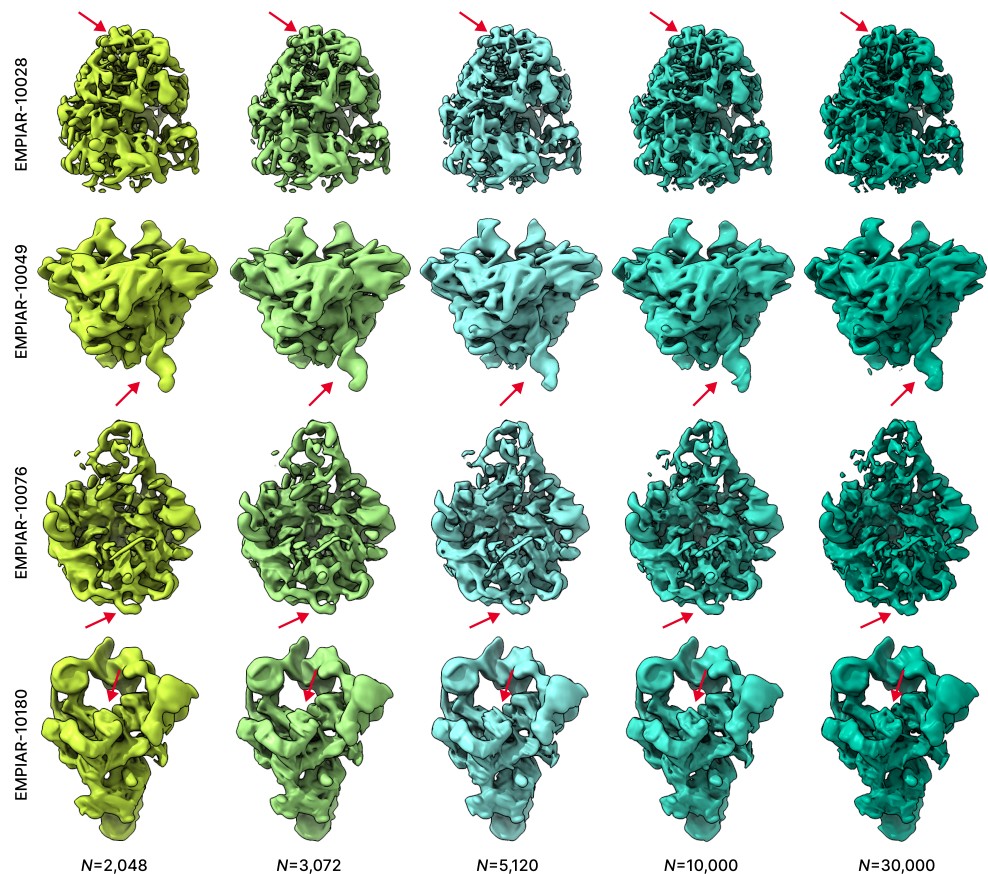

Figure 6: Qualitative evaluation of reconstruction performance with different numbers of Gaussians. Increasing the number of Gaussians leads to visibly improved reconstructions, with finer structural details and enhanced sharpness. Red arrows mark representative regions that highlight the qualitative differences for clearer comparison across settings.

We present visual comparisons of reconstruction results using different numbers of Gaussians. As shown in Fig. 6, increasing the number of components yields progressively sharper and more detailed structures. These qualitative observations align with the quantitative improvements in FSC curves reported in Fig. 3. Red arrows highlight representative regions where the differences in reconstruction quality are especially pronounced, facilitating direct visual comparison across settings.

## D.2    ISOTROPIC VS. ANISOTROPIC

CryoSplat represents 3D volumes using anisotropic Gaussians while remaining fully compatible with the isotropic formulation widely adopted in prior works (Chen & Ludtke, 2021; Chen et al., 2023a;b; Schwab et al., 2024; Chen, 2025). When the scaling is isotropic, i.e., $s_x = s_y = s_z = \sigma$,

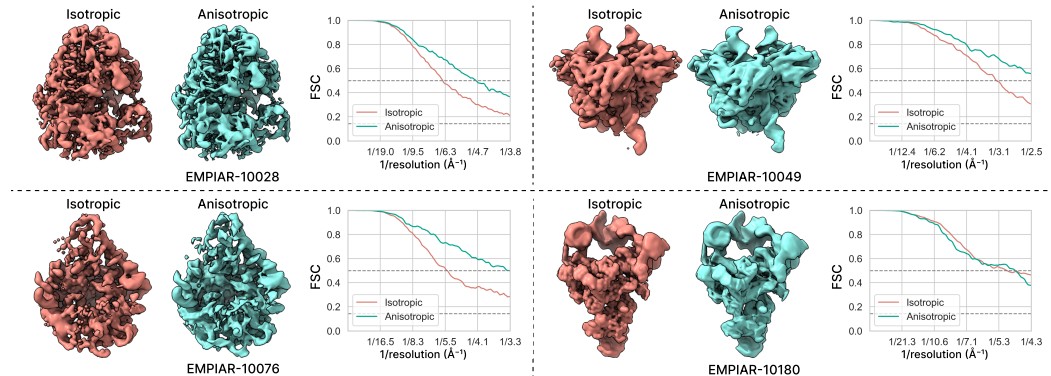

Figure 7: Qualitative and quantitative comparison of isotropic and anisotropic GMMs ($N = 30{,}000$) on four real datasets. FSC curves show that anisotropic Gaussians consistently achieve higher correlations across spatial frequencies, indicating improved reconstruction accuracy. Volume visualizations further reveal that anisotropic GMMs better recover fine structural details and elongated features, whereas isotropic Gaussians tend to fragment such regions.

the anisotropic Gaussian exactly reduces to the standard isotropic form:

$$G(\boldsymbol{r}|\boldsymbol{\mu}, \sigma) = \frac{1}{(2\pi)^{\frac{3}{2}}\sigma^3}\exp(-\frac{\|\boldsymbol{r} - \boldsymbol{\mu}\|_2^2}{2\sigma^2}), \tag{22}$$

allowing direct integration into existing isotropic GMM-based pipelines.

We investigate the impact of isotropic versus anisotropic Gaussians on reconstruction quality. As shown in Fig. 7, anisotropic GMMs achieve higher FSC scores across spatial frequencies and produce sharper, more detailed structures. Subjectively, isotropic Gaussians struggle to capture elongated features and are often captured by noise, which may contribute to the unstable convergence from random initialization reported in previous methods. These results highlight the improved representational capacity and reconstruction robustness enabled by anisotropic modeling.

## D.3 SIGNAL-TO-NOISE RATIO

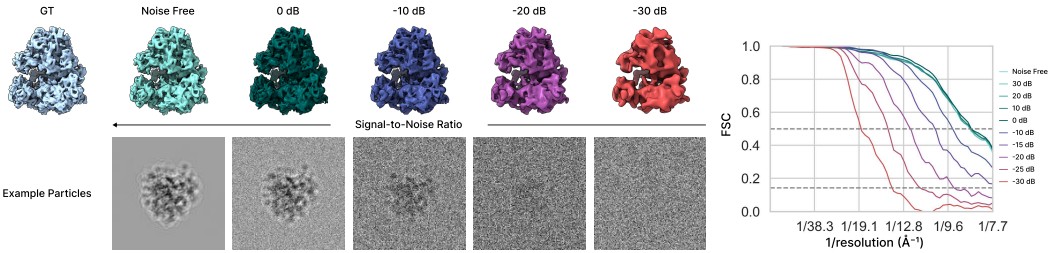

Figure 8: Reconstruction performance under varying SNRs. **(top)** Ground-truth (GT) and reconstructed volumes at different SNR levels. **(bottom)** Example synthetic particle images corresponding to each SNR. **(right)** FSC curves between GT and reconstructed volumes across SNRs.

We study the effect of SNR levels on cryoSplat with $5{,}120$ Gaussians using the synthetic 80S dataset described in Appendix B. Figure 8 shows example synthetic particles, reconstructed volumes, and FSC curves under varying SNRs. FSCs are computed between the ground truth (GT) and reconstructed volumes. Overall, cryoSplat shows strong noise robustness: SNRs above $0$ dB have little impact on reconstruction; high resolution is preserved even under severe noise at $-15$ dB, and reconstructions remain satisfactory at $-20$ dB, despite particles being barely visible.

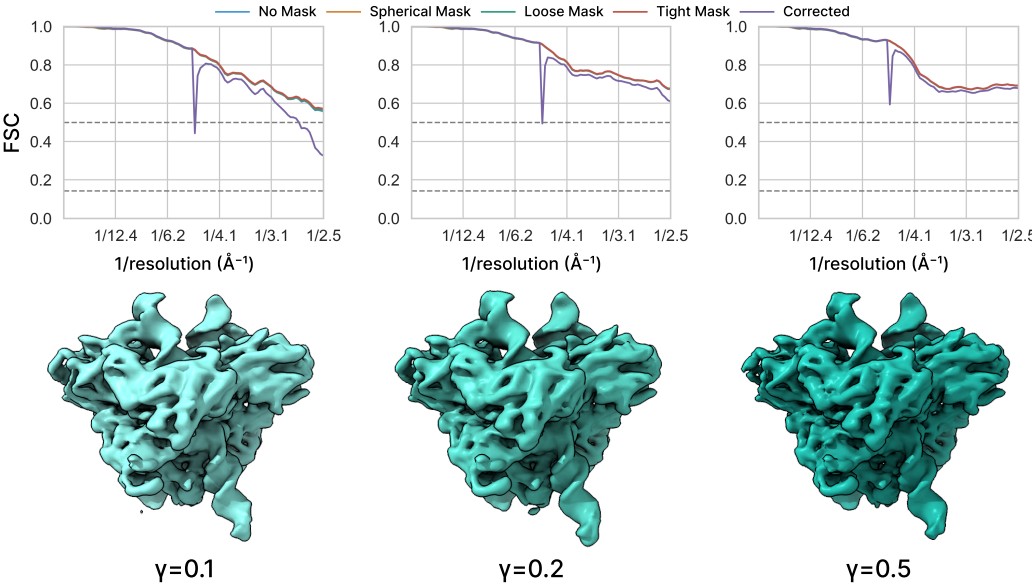

Figure 9: Reconstruction using 30,000 Gaussians under different exponential decay parameters $\gamma$. For each setting, the half-map FSC, masked FSC, and corrected FSC curves are plotted together for better comparison. When $\gamma = 0.1$, the half-map FSC exceeds the corrected FSC, indicating an overestimation of resolution due to strong self-consistency. Increasing $\gamma$ reduces this discrepancy, and at $\gamma = 0.5$ the corrected FSC closely matches the half-map FSC, suggesting that no detectable artificial bias is introduced.

## D.4  CORRECTED FSC

Half-map FSC is fundamentally a measure of self-consistency rather than visible structural detail. It is interpreted as a proxy for resolution under the assumption that (i) the SNR decreases at high frequencies, and therefore (ii) two independently reconstructed half maps should lose consistency in the high-resolution regime. If a method maintains strong self-consistency even under low SNR, whether due to genuine robustness or to an inherent bias, the half-map FSC may overestimate the true resolution. For example, if a method consistently overfits random noise into reproducible artificial patterns, the two half maps may show spurious agreement.

Corrected FSC (Chen et al., 2013) is specifically designed to detect such artificial bias. It does so by randomizing Fourier phases: half maps with randomized phases should share no meaningful consistency. Any remaining agreement is therefore interpreted as bias and subtracted from the FSC curve. In our results shown in Fig. 9, we indeed observe a discrepancy between the half-map FSC and the corrected FSC, indicating that the half-map FSC tends to overestimate cryoSplat's resolution. Importantly, however, this discrepancy can be removed by slightly increasing the exponential decay parameter $\gamma$. When $\gamma = 0.5$, the corrected FSC closely follows the original half-map FSC, suggesting that no detectable artificial bias is introduced by cryoSplat.

We also observe that FSC curves computed under different masking levels remain tightly aligned. This indicates that cryoSplat suppresses noise effectively outside the signal-support region: the noise level is so low that applying a mask has virtually no effect on the FSC, consistent with our synthetic-data experiment in Appendix D.3, showing the strong denoising capability of GMM-based representations.

Having ruled out detectable artificial bias via corrected FSC, we next analyze why the half-map FSC may still overestimate the resolution for cryoSplat. We attribute this to the GMM's strong ability to maintain self-consistency during optimization. This property is largely driven by the inherently low-pass nature of Gaussian kernels. On the one hand, the low-pass behavior encourages the model to fit low-frequency components more readily, yielding smoother volumes. As we show later in Appendix D.5, this effect can be substantially mitigated by increasing the number of Gaussians, which restores high-frequency detail. On the other hand, the same low-pass property also

contributes to excellent self-consistency: both quantitative metrics and qualitative assessment show that this consistency does not manifest as harmful artifacts. However, because the self-consistency is exceptionally strong, conventional FSC-based resolution estimation can become overly optimistic. As a result, qualitative assessment and domain-expert evaluation remain the most trustworthy way to evaluate the effective resolution produced by cryoSplat. Developing rigorous and objective quality metrics tailored to GMM-based cryo-EM reconstruction remains an important open question.

### D.5 RECONSTRUCTION BEHAVIOR IN THE ULTRA-HIGH GAUSSIAN COUNT REGIME

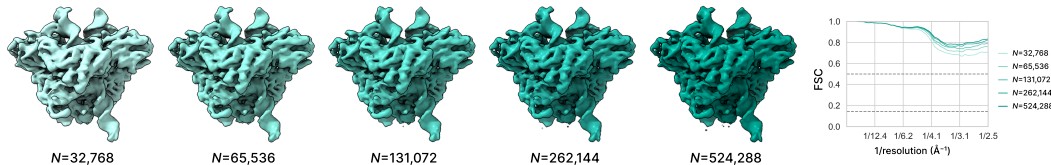

Figure 10: Qualitative and quantitative evaluation of reconstruction performance with ultra-high Gaussian counts. (**Left**) Reconstructed 3D volumes. (**Right**) FSC curves are plotted for quantitative evaluation. Gray dashed lines indicate the standard resolution thresholds of $0.5$ and $0.143$.

To further examine the representational capacity and optimization behavior of GMM-based representations, we increase the Gaussian count from $32{,}768$ up to $524{,}288$, as shown in Fig. 10. To ensure that the increased capacity can be fully utilized, we raise the exponential decay parameter to $\gamma = 0.5$ and train for 10 epochs.

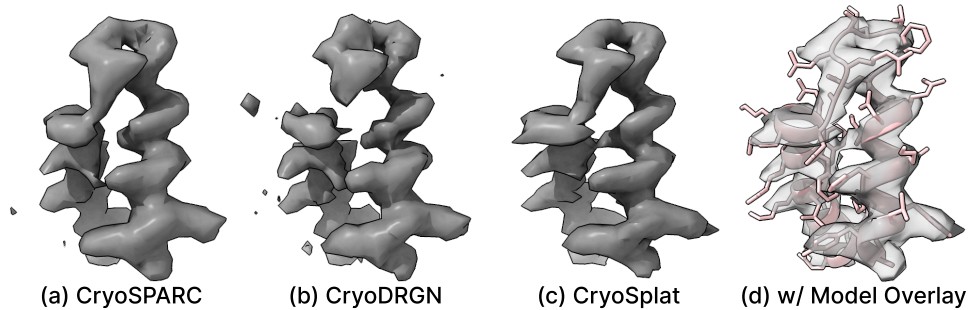

(a) CryoSPARC     (b) CryoDRGN     (c) CryoSplat     (d) w/ Model Overlay

Figure 11: Feature comparison at an $\alpha$-helix in the RAG1–RAG2 core region cropped from the reconstructed volume. CryoSplat with $524{,}288$ Gaussians produces a sharp and continuous helical density that is comparable to cryoSPARC, while cryoDRGN shows breaks along the backbone and retains visible noise. Overlay with the atomic model demonstrates that cryoSplat accurately recovers the backbone trace and resolves larger side-chain features, including aromatic rings.

Increasing the number of Gaussians consistently improves both quantitative and qualitative reconstruction quality. We observe monotonic increases in FSC scores as the Gaussian count grows, together with visibly sharper structural details. At the highest resolution regime (524k Gaussians), many fine-scale features such as $\alpha$-helices become clearly resolved, as shown in Fig. 11.

This is surprising given the common expectation that larger models are harder to optimize and more prone to unstable convergence. Instead, the opposite effect is observed: larger GMMs exhibit better consistency between half-maps and converge to higher FSC, indicating more stable optimization dynamics in the ultra-high-capacity setting. This suggests that additional Gaussians provide finer local modeling flexibility, allowing the representation to better accommodate noise and subtle structural variability without overfitting.

Finally, even at $524{,}288$ Gaussians, the model remains substantially more compact than voxel-based grids, since $524{,}288 \times 11 < 256^3$. However, this ultra-high-Gaussian regime currently incurs a significantly increased computational cost, making it impractical for routine use. We view these results

### D.6 RECONSTRUCTION BEHAVIOR IN THE LOW GAUSSIAN COUNT REGIME

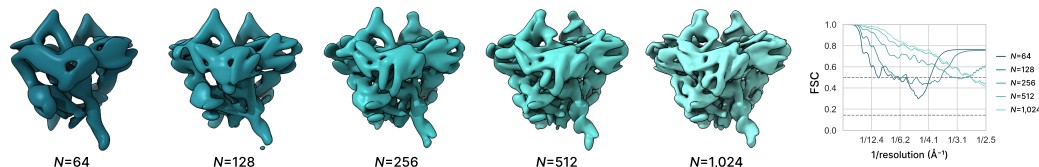

Figure 12: Qualitative and quantitative evaluation of reconstruction performance with low Gaussian counts. (**Left**) Reconstructed 3D volumes. (**Right**) FSC curves are plotted for quantitative evaluation. Gray dashed lines indicate the standard resolution thresholds of $0.5$ and $0.143$.

We additionally investigate how GMM-based reconstructions behave when the number of Gaussians is severely limited, reducing the count from $1{,}024$ down to $64$, as shown in Fig. 12. As the Gaussian count decreases, both quantitative and qualitative reconstruction quality degrade consistently. The reconstructed densities become progressively blurred and blob-like, and once the count falls below $1{,}024$, the maps begin to exhibit clearly visible Gaussian ellipsoids in place of coherent structural details. This reflects the insufficient spatial degrees of freedom available to represent localized structural details. The FSC curves also reveal a distinctive failure pattern when the number of Gaussians becomes extremely small. Instead of exhibiting a smooth decay, the curves dip at intermediate frequencies and then rise again at high frequencies. This rise does not reflect genuine high-frequency agreement; it occurs because the high-frequency components are largely absent, and the Fourier amplitudes of both half-maps approach zero in these bands, which leads to unreliable consistency and inflated FSC values. Such low-Gaussian-count configurations are not practical for real reconstruction tasks. Although computational cost is reduced, the representation becomes too under-resolved to yield reliable maps, and both qualitative appearance and quantitative metrics lose interpretability.

### D.7 ABLATION ON INITIALIZATION STRATEGY

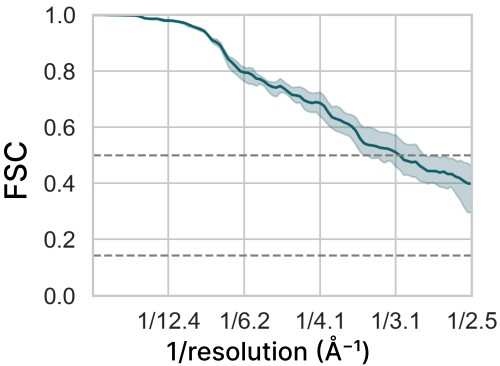

Figure 13: Using $2{,}048$ Gaussians, the RAG1–RAG2 complex (EMPIAR-10049) is reconstructed under $10$ random seeds ($0$–$9$). The figure shows the mean half-map FSC with the minimum–maximum envelope across these runs. The narrow band indicates that cryoSplat converges to highly consistent results across initializations, demonstrating strong robustness and stability.

We also examine how sensitive the method is to the choice of initialization. First, we fix the GMM configuration and only vary the random seed from $0$ to $9$. For each run, we compute the half-map FSC curve and then aggregate the results into a mean curve with an upper and lower envelope over all seeds. As is shown in Fig. 13, the envelopes are tight across all frequencies, indicating that both convergence behavior and final result are highly robust to random seed choices.

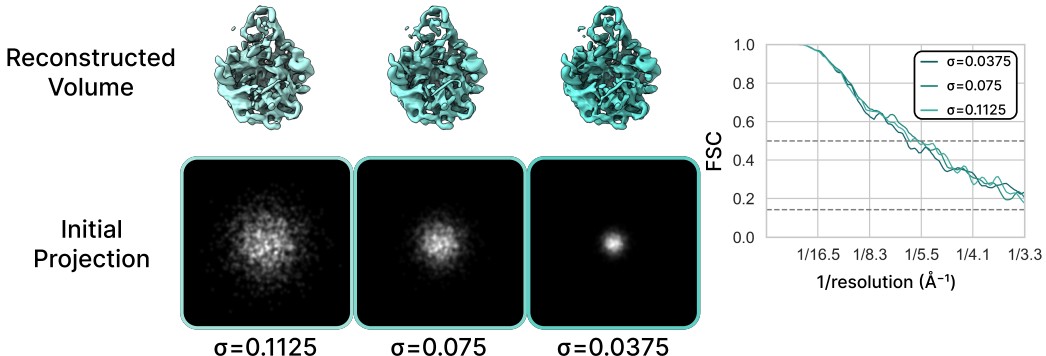

Figure 14: Initialization ablation on EMPIAR-10076 using $2{,}048$ Gaussians. Reconstructed volumes (top-left), initial Gaussian projections (bottom-left), and FSC curves (right) under three initialization spreads $\sigma \in \{0.0375, 0.075, 0.1125\}$. Both visual reconstructions and FSCs remain nearly identical across all settings, indicating that cryoSplat is highly robust to the choice of initialization.

A more challenging setting arises when the underlying structure exhibits substantial heterogeneity, introducing ambiguity during early optimization. EMPIAR-10076 is a representative example, containing pronounced compositional variability across the complex. To evaluate whether such variability affects the stability of the initialization, we vary the spatial spread of the initial Gaussian locations by sampling with $\sigma \in \{0.0375, 0.075, 0.1125\}$, which approximately correspond to placing Gaussians within spherical regions of radii $3E/2$, $E/2$, and $E/4$, respectively. The resulting initial projections under these settings, shown in Fig. 14, clearly illustrate the differences. Notably, the smallest spread ($\sigma = 0.0375$) does not cover the full region where the signal is present and therefore represents a substantially under-dispersed initialization. Despite this, all configurations converge to nearly identical reconstructions on this heterogeneous dataset: both the FSC curves and the visualized volumes are highly consistent. This indicates that even in cases with significant variability and ambiguous starting configurations, the optimization remains robust to the choice of initialization.

## E    THE USE OF LARGE LANGUAGE MODELS

We acknowledge GPT-5 for its assistance with grammar correction, sentence shortening, and language polishing. No part of the research design, analysis, or conclusions was generated by LLMs.

