# OpenReview forum: "CryoSplat: Gaussian Splatting for Cryo-EM Homogeneous Reconstruction"
_ICLR.cc/2026/Conference — ICLR 2026 Poster_

### Official Review · Reviewer_LabF · 2025-10-25

**Soundness:** 3
**Presentation:** 4
**Contribution:** 2
**Rating:** 2
**Confidence:** 4

**Summary:**

Single particle cryo electron microscopy (cryo-EM) is a field whose main goal is to reconstruct the 3D shape of molecules from thousands of transmission electron microscope (TEM) images. This field has experienced tremendous progress over the last decade, including several mature software packages for 3D reconstruction such as RELION [1], cryoSPARC [2] and CryoDRGN [3]. The reconstruction pipeline for cryo-EM is comprised of several different steps: particle picking (identifying the location of the molecules in the noisy images), estimation of the point-spread function and ab-initio reconstruction (obtaining an initial low-res reconstruction). This is typically followed by the estimation of viewing angles combined with high-resolution 3D refinement. This paper is focused on the last step: the high-resolution 3D reconstruction of a single (fixed) molecule with known viewing angles. It proposes a simple approach inspired by 3D Gaussian splatting [4] that the authors name CryoSplat. It is a more-or-less direct adaptation of 3D splatting from computer graphics to the cryo-EM domain, making the following changes to match the TEM imaging modality:
1. The projection is orthogonal, in contrast to the perspective projection (pinhole camera) model typically used in computer graphics.
2. There are no occlusions. The Gaussians are simply added together. So there's no need for a z-buffer and alpha-blending.
3. The algorithm does not need to figure out which Gaussians are outside the camera's view since the entire molecule is always in view. The authors still divide the image into tiles, but this is merely an implementation detail for better GPU performance.
4. The projection image is convolved with a point-spread function.

The main contribution of this paper is a demonstration that, given input images with matching projection orientations, a high-resolution 3D GMM model can be used to reconstruct the 3D volume using gradient-based optimization.

[1] Scheres, Sjors HW. "RELION: implementation of a Bayesian approach to cryo-EM structure determination." Journal of structural biology 180, no. 3 (2012): 519-530.

[2] Punjani, Ali, John L. Rubinstein, David J. Fleet, and Marcus A. Brubaker. "cryoSPARC: algorithms for rapid unsupervised cryo-EM structure determination." Nature methods 14, no. 3 (2017): 290-296.

[3] Zhong, Ellen D., Tristan Bepler, Bonnie Berger, and Joseph H. Davis. "CryoDRGN: reconstruction of heterogeneous cryo-EM structures using neural networks." Nature methods 18, no. 2 (2021): 176-185.

[4] Kerbl, Bernhard, Georgios Kopanas, Thomas Leimkühler, and George Drettakis. "3D Gaussian splatting for real-time radiance field rendering." ACM Trans. Graph. 42, no. 4 (2023): 139-1.

**Strengths:**

* I like the simplicity of the approach. The authors have demonstrated that one can obtain high-resolution reconstructions simply by running backprop. I am not aware of previous literature that demonstrated this.

* The paper is very clearly written.

**Weaknesses:**

* The benchmarks do not compare the method to leading packages for homogeneous reconstruction but only to CryoDRGN (and a backprojection baseline). While CryoDRGN is a leading package for heterogeneous reconstruction, it is not the best at homogeneous reconstruction. At a minimum, I do not think the paper can be accepted without a comparison to a leading homogeneous reconstruction method such as RELION or cryoSPARC.

* The novelty of this work is not very high in my opinion. Gaussian mixture models have been used by several authors in cryo-EM reconstruction (e.g. [5],[6]) but mostly to attack the much more difficult problem of heterogeneous reconstruction, where molecules have continuous degrees of freedom. This is probably due to the fact that in cryo-EM the homogeneous reconstruction problem is considered to be "solved". The authors acknowledge the works of Chen et al. and others, but claim that they are different from these works because they do not rely on an approximate 3D model of the molecule ("consensus model") for initialization. Indeed, CryoSplat does not require such an initial model, but it takes the orientations of the images as input.  The issue is that on experimental data the orientations are typically obtained by rotating a consensus model and comparing the resulting projections to the input images... So a consensus model is still used in the pipeline. As far as I can tell, the optimization is a straightforward application of backpropagation with no "secret sauce". Hence, the main contribution of the paper is a demonstration that backprop on a GMM with known orientations can result in high resolution models. The contribution would be much more significant if the method was able to also estimate the viewing angles from scratch.

* This submission seems to have significant overlap to GEM (ICLR submission 6298) even with regards to the choice of datasets. While not a weakness, this is something that the chairs need to take into consideration. In particular, it would not make sense to accept both papers to the same conference.

* The code was not included in the submission so I could not review it.

[5] Chen, Muyuan, Michael F. Schmid, and Wah Chiu. "Improving resolution and resolvability of single-particle cryoEM structures using Gaussian mixture models." Nature methods 21, no. 1 (2024): 37-40.

[6] Shekarforoush, Shayan, David B. Lindell, Marcus A. Brubaker, and David J. Fleet. "Reconstructing Heterogeneous Biomolecules via Hierarchical Gaussian Mixtures and Part Discovery." arXiv preprint arXiv:2506.09063 (2025).

**Questions:**

* Is there any "secret sauce" in the implementation that I missed? i.e. non-standard choices in the optimization problem definition or implementation that are critical to getting the method to converge or to obtain high-resolution 3D volumes.

* Why did you not include the code in the submission? Are you willing to attach an (anonymized) zip file during the review cycle?

Minor comments (not questions).
* In Figure 1 the black and orange lines should probably have arrowheads instead of ball heads to clearly show the direction.
* The last paragraph of Section 3.3.4 seems out of place.

---

> ### Author Response · Authors · 2025-11-21
>
> ### **1. CryoSPARC**
> Thank you for the suggestion. In response, we have replaced the previously shown back-projection reconstruction with the corresponding CryoSPARC volume to ensure a fair visual comparison. For the runtime and memory evaluations, however, we continue to rely on cryoDRGN’s standalone back-projection script. Highly integrated packages such as CryoSPARC or RELION launch multiple auxiliary processes and system-level tasks, introducing additional overhead that is unrelated to the reconstruction algorithm itself. Consequently, their reported runtime and memory usage do not reflect the intrinsic computational cost. Using a clean back-projection implementation allows us to isolate and measure the true algorithmic efficiency.
>
> ### **2. Novelty**
> As discussed in Sec. 2.1.3, GMM-based approaches are typically regarded as unstable and often require special initialization schemes or additional constraints to remain trainable. Similarly, 3DGS for novel view synthesis usually depends on point cloud initialization. Our main contribution is to demonstrate that by carefully aligning the Gaussian splatting framework with the physics of cryo-EM imaging, a GMM can naturally converge in a stable manner without such requirements. Our experiments in Sec. E.5 and E.9 further show that the method maintains strong self-consistency even on extremely noisy data, which is a desirable property for ab initio reconstruction.
>
> We would also like to note that this interpretation of the contribution is consistent with another reviewer’s assessment, who described “*the integration of Gaussian splatting with cryogenic electron microscopy*” as “*novel and promising*.”
>
> ### **3. Code Implementation**
> We have uploaded a demo of our implementation in supplementary material. The code is fully integrated into the CryoDRGN framework, with no hidden or proprietary modifications. Data preprocessing and postprocessing strictly follow CryoDRGN’s pipeline, including pose handling, CTF correction, and half-map FSC computation. This demo can easily reproduce the main results in our paper. The full version will be released upon publication.

---

> ### Author Response · Authors · 2025-11-27
> **Follow-up Comment for Reviewer LabF**
>
> Thank you again for your thoughtful review and for the time you have dedicated to evaluating our submission. We sincerely appreciate your comments. We have provided detailed, point-by-point responses to all of your concerns, including the cryoSPARC baseline, the question of technical novelty, and reproducibility and code availability. We warmly invite you to take a look whenever convenient, and we would be grateful for any additional feedback you may have.
>
> Regarding the concern about whether the problem addressed in our paper has already been solved, we would like to offer a brief clarification. **A very recent NeurIPS 2025 accepted paper (Shekarforoush et al., 2025)** on GMM-based cryo-EM reconstruction presents two observations that **are consistent with the challenges discussed in our work**.
>
> First, the paper states that *“the optimization dynamics are highly sensitive to the initial values”* (Appendix C), which supports our statement in Line 131 that *“Without such guidance, random initialization leads to unstable optimization and poor reconstruction quality.”*
>
> Second, the paper notes that *“Given the input density map… Gaussians are then seeded at those positions.”* (Appendix C). This is consistent with our statement in Line 64 that existing methods typically rely on consensus volumes or atomic models for initialization.
>
> Although GMMs have been explored in cryo-EM since E2GMM in 2021, this very recent NeurIPS work (Shekarforoush et al., 2025) still requires specialized initialization. This indicates that **stability under unconstrained random initialization has remained a challenging problem for GMM-based cryo-EM methods.** We hope this additional context helps clarify the motivation and contribution of our work.
>
> Thank you again for your time and thoughtful evaluation. We sincerely appreciate your effort, and we would welcome any further comments you may have.

---

### Official Review · Reviewer_qU3z · 2025-10-28

**Soundness:** 3
**Presentation:** 2
**Contribution:** 3
**Rating:** 6
**Confidence:** 4

**Summary:**

The paper adapts 3D Gaussian Splatting to homogeneous reconstruction in cryo-EM, using only raw picked particle images and assumed known poses. CryoSplat generally exceeds the resolution (measured via Fourier Shell Correlation) of voxel-space backprojection and CryoDRGN, on representative proteins.

**Strengths:**

The method is a natural combination of 3D Gaussian Splatting, which works well in computer vision, to cryo-EM homogeneous reconstruction. It includes some adaptations necessary for the forward model and high noise of cryo-EM, and quantitative results (FSC) are encouraging.

**Weaknesses:**

There are some aspects of the presentation that could be improved.
- The bullet point list of contributions at the end of the introduction slightly overstates the contributions, in my view. In particular, the efficient CUDA implementation would seem to be largely based on that of 3DGS, with fairly minor modifications. This is still a contribution to be sure, but the contribution description should acknowledge if this code is largely based on 3DGS code.
- Some of the descriptions of voxel-based methods strike me as a bit unfair/unsupported. For example, line 109 describes these methods as “inherently discrete”--this is not the case when voxels use interpolation beyond nearest neighbor. It is ironic that this is described as a limitation of voxels, when (although Gaussians are technically infinitely-supported) 3DGS crops the spatial extent of each Gaussian and is thus more discrete of a representation than a grid with interpolation, since there is no interpolation between distinct Gaussians. Line 124 describes Gaussian mixture models as supporting differentiable optimization; this is only the case when the Gaussians are modeled with their full support, and even so the Gaussian and its gradient are negligible over much of its support, making the gradients of limited value beyond the main ellipsoid of support. Additionally, section 4.3 describes neural representations and Gaussians as more flexible for heterogeneous reconstruction compared to voxel grids. This claim is unsupported (e.g. it is straightforward to imagine combining a canonical 3D model in the form of a voxel grid with a neural or explicit deformation/conformation model, just as one could for a neural or Gaussian representation).
- The last paragraph on page 3 describes three aspects of the original 3DGS that must be adapted for application to cryo-EM. The first is the switch from perspective to orthographic, which is clearly described. However, the second and third aspects are not clearly described. Aspect (ii) is described as prioritizing photorealism over physical accuracy, and aspect (iii) is described as misaligned coordinates with respect to the Fourier slice theorem. Aspect (ii) is too vague; aspect (iii) makes it sound like the Fourier slice theorem is used in the forward model, but this is not the case since the forward model here is applied in real-space, so it’s not clear why misaligned coordinates would be an issue. Is this because the contrast transfer function is applied in Fourier domain?
- In Figure 2, the resolution is described as being calculated based on the point where the FSC curve crosses 0.143. However, the graphs of FSC on the right of the figure seem to have arbitrary x axis scales, that do not always show where FSC for CryoSplat crosses 0.143. Moreover, the resolutions reported for CryoSplat do not seem to match these crossing points.
- In Figure 2, I acknowledge that CryoSplat achieves higher FSC. However, I am confused by this because visually in the ChimeraX figures the reconstructions from CryoSplat appear smoother than those from both backprojection and CryoDRGN, and in some cases (e.g. the second row) there are fine details in which backprojection and CryoDRGN are in agreement with each other, but the same features are not present in CryoSplat. I am not sure how to square this observation with the quantitatively higher FSC for CryoSplat, but I am concerned of a potential bug in computing FSC.
- Figure 3 caption claims that 10,000 Gaussians are usually sufficient, yet in all four subfigures there is meaningful improvement moving from 10,000 Gaussians to 30,000. Moreover, in Figure 2 CryoSplat uses 30,000 Gaussians. The goal should not be just to outperform baselines, but to maximize resolution and stability of reconstruction.
- The initialization strategy for the Gaussians is described on line 396, but the rationale behind it is not explained until the appendix. The rationale is more important than the numbers themselves.
- Section 4.2 describes some reconstructions as producing artifacts, and points out the location of artifacts in Figure 2. How can you know what is an artifact, when you do not have a ground truth volume for comparison?

**Questions:**

Please refer to questions embedded in weaknesses. I am open to raising my score if these presentation concerns are addressed.

---

> ### Author Response · Authors · 2025-11-21
>
> ### **1. Overstated Contribution**
> Our CUDA implementation builds on the tile-based 3DGS framework, but both gradients and forward operators were re-derived under the cryo-EM imaging model (Appendix A.2). While some low-level computational structure is retained, these derivations require new CUDA kernels and substantially change the mathematical content of the renderer. We revised the contribution statement to make this clearer.
>
> ### **2. Unsupported Limitation of Voxel Grids**
> Thank you for pointing out the concerns. The statement that voxel grids are “inherently discrete” has been removed. While voxel grids can use high-order interpolation, we clarify the reviewer’s point about GMM differentiability: at high resolution, Gaussians shrink spatially but expand in Fourier support and are strongly modulated by CTF. As shown in Fig. 1 through the simulated projection example, this interaction spreads the effective support of each Gaussian in real space, allowing gradients from much larger spatial regions than their visible ellipsoids.
>
> Regarding heterogeneous reconstruction, voxel grids are theoretically compatible. However, VAEs require decoding full volumes; for a box of 128, this entails larger than 2 million output parameters, resulting in extremely high complexity and unstable training. Gaussian parameterizations are compact and thus more suitable for latent-conditioned heterogeneous models, which explains their prevalence.
>
> ### **3. Clarification**
> Thank you for the helpful clarification request. The description at line 161 has been revised. For aspect (iii), although our renderer operates in real space, the Fourier slice theorem remains central: cryo-EM imaging follows Fourier optics, and CTF is applied in the Fourier domain. Therefore, projection coordinates must be aligned with the FFT grid; Appendix Fig. 5 illustrates this in detail.
>
> ### **4. Varying X-Axis Scale**
> The FSC x-axis differs across datasets because pixel size and hence the physical Nyquist frequency vary, following conventions in cryoDRGN and E2GMM.
>
> ### **5. Visualization Does Not Match FSC**
> Half-map FSC is a measure of self-consistency rather than visible detail. Resolution is measured intermediately under the assumption that SNR is lower and it is harder for half maps to keep their consistency at high resolution. However, if a method can keep self-consistency under low SNR or has some inherent bias, such intermediate measure may overestimate the resolution. Corrected FSC is designed to detect and eliminate potential bias inside a method through randomizing the phase. However, experiment shows that corrected FSC fails to detect salient bias from GMMs with larger Gaussian counts, suggesting that the overestimated resolution is mainly due to GMMs' good self-consistency. Thus, the current practical way to evaluate GMM-reconstructed maps is visually recognizing the structural details. How to quantitatively measure the resolution of GMM-reconstructed maps is still an open question. We will keep exploring a better solution. For more discussions, please refer to Sec. E.6.
>
> ### **6. Potential Bugs**
> Thank you for raising the concern regarding possible implementation bugs. To rule out implementation issues, we provide an anonymous demo where our renderer is integrated into cryoDRGN, with preprocessing and FSC computation handled entirely by cryoDRGN's pipeline. The demo reproduces our reported results, indicating that the observed behavior is not due to implementation errors. The full demo code is included in the supplementary materials.
>
> ### **7. "10,000" is Sufficient**
> We clarified that the claim is based on FSC measurements. The number of required Gaussians is task-dependent: ab initio reconstruction mainly needs pose recovery, for which ~2k Gaussians suffice; heterogeneous reconstruction primarily requires separating compositional and conformational states, where ~10k is adequate; high-resolution homogeneous refinement may require substantially more.
>
> ### **8. Rationale of Initialization**
> Thank you for the suggestion. We have added a more detailed rationale for our initialization strategy in Sec. 4.1 of the implementation details.
>
> ### **9. How to Define Artifact**
> Since no ground truth exists for experimental data, “artifact” refers to visually implausible density—isolated speckles, discontinuous fragments, or spurious high-frequency blobs inconsistent with molecular geometry. To avoid ambiguity, we now use “density fragments” or “spurious spikes” in Sec. 4.2.

---

> > ### Comment · Reviewer_qU3z · 2025-11-23
> >
> > Thanks for the detailed responses to my questions and comments. I remain in favor of acceptance. Regarding mismatch between metrics and visual results, I would love to see a more robust metric than FSC (robust in the sense that a method with a specific bias could achieve high FSC even without actually achieving higher physically-accurate resolution), but I am not aware of such a metric in the absence of ground truth, so I cannot fault the authors for using an (albeit flawed) but standard metric.

---

### Official Review · Reviewer_Mi5f · 2025-10-31

**Soundness:** 3
**Presentation:** 3
**Contribution:** 3
**Rating:** 6
**Confidence:** 4

**Summary:**

CryoSplat introduces a 3DGS-based homogeneous cryo-EM reconstruction framework that integrates orthographic Gaussian splatting with the physics of electron imaging. The method derives a projection model with view-dependent normalization and FFT-aligned coordinates so that real-space rendering remains consistent with Fourier-based acquisition. A CUDA-accelerated rasterizer supports tens of thousands of anisotropic Gaussians optimized from random initialization using a simple MSE loss. One advantage of cryoSplat comparing to the previous methods such as E2GMM is cryoSplat can converge from random initialization without the need of the traditional consensus reconstruction. Extensive experiments have conducted on EMPIAR-10028, 10049, 10076, and 10180 with provided poses and CTFs, cryoSplat consistently surpasses backprojection and CryoDRGN in FSC resolution, reaching 2.49 Å on EMPIAR-10049.

**Strengths:**

1. The paper adapts Gaussian splatting to cryo-EM with a physically grounded orthographic projection, view-dependent normalization, and FFT-aligned coordinates, enabling stable reconstruction from random initial Gaussians without external priors.
2. The CUDA-based renderer and simple MSE objective deliver practical efficiency, achieving 2–3× higher rendering throughput than CryoDRGN, convergence in five epochs, and support for up to 30k Gaussians.
3. Experiments on four EMPIAR datasets show consistent FSC gains over backprojection and CryoDRGN—including a 2.49 Å resolution on EMPIAR-10049, and the ablations intelligently cover both fidelity and runtime, showing that larger Gaussian sets improve accuracy while incurring longer training and rendering.
4. The exposition is clear and well-motivated, the paper is well-written and easy to follow.

**Weaknesses:**

1. Even with 30k Gaussians, the reconstructions in Fig. 2 remain noticeably smoother than the backprojection or CryoDRGN baselines, raising the concern that the strong Gaussian prior could yield high FSC scores while underfitting finer details. Please explore much larger mixtures—e.g., 50k or 100k components—to see whether the qualitative sharpness catches up and whether the FSC improvements persist or saturate.
2. The Gaussian-count ablation suggests almost linear gains in FSC as N increases, without clear signs of convergence, so it is unclear how the model behaves in the low-capacity regime. A complementary study reducing N well below 2,048 would clarify whether the FSC metric stays artificially high despite visibly degraded reconstructions, helping disentangle genuine data fit from prior-driven smoothing.
3. The current evaluation remains limited to homogeneous reconstruction and compares primarily against CryoDRGN, which targets continuous heterogeneity; adding a popular cryoSPARC baseline with the same pose inputs, and outlining how cryoSplat could extend beyond static cases would make the empirical story more compelling.

**Questions:**

Addressing the following points, which is aligned with the weaknesses above, would help me reassess the paper more favorably.

1. Provide qualitative comparisons or additional reconstructions (e.g., with 50k–100k Gaussians) to check whether sharper details emerge and the FSC gains remain trustworthy.
2. Extend the Gaussian-count ablation to much smaller mixtures so we can see whether the FSC metric genuinely tracks reconstruction quality when capacity drops.
3. Include a cryoSPARC baseline that uses the same pose inputs, and discuss about the future direction for adapting cryoSplat to heterogeneous or dynamic reconstruction.

---

> ### Author Response · Authors · 2025-11-21
>
> ### **1. More Gaussians**
>
> As shown in Sec. E.7, increasing the number of Gaussians from 32,768 up to 524,288 consistently improves both quantitative and qualitative reconstruction quality. FSC curves rise monotonically with model capacity, and the visual maps exhibit progressively sharper structural details. At the highest setting (524k Gaussians), many fine-scale features, including α-helices and even DNA double-strands, become clearly resolved. These results confirm that additional Gaussians provide meaningful expressive power and lead to substantial reconstruction improvements.
>
> ### **2. Less Gaussians**
>
> As detailed in Sec. E.8, we also evaluate the behavior of the method when the number of Gaussians is severely restricted, reducing the count from 1,024 down to 64. In this low-capacity regime, both visual quality and FSC degrade consistently: the reconstructed densities become increasingly blurred, and below 1,024 Gaussians the maps begin to show clear Gaussian ellipsoids instead of coherent structural features. The FSC curves exhibit a characteristic failure mode in which mid-frequency values drop while high-frequency values rise artifactually due to vanishing Fourier amplitudes. These results confirm that extremely small Gaussian counts provide insufficient spatial degrees of freedom for accurate reconstruction, leading to both qualitative and quantitative failure.
>
> ### **3. CryoSPARC**
>
> Thank you for the suggestion. Following the comment, we have replaced the previously reported back-projection reconstruction with the corresponding CryoSPARC result to ensure a fair comparison. However, for the runtime and memory experiments, we continue to use the cryoDRGN's back-projection script. Highly integrated software such as CryoSPARC or RELION invokes multiple auxiliary processes and system-level operations, which introduce additional overhead unrelated to the reconstruction algorithm itself. As a result, measuring runtime or memory usage through these platforms would not provide a fair or interpretable comparison. Using a clean back-projection implementation allows us to isolate the computational cost of the reconstruction algorithm itself.
>
> ### **4. Extension to Heterogeneous Reconstruction**
>
> We agree that extending our homogeneous framework to heterogeneous reconstruction is an important next step. A widely used approach is to incorporate a VAE, where each particle is embedded into a latent space and the decoder outputs particle specific Gaussian parameters. This enables the GMM to vary smoothly across different conformations.
>
> Recent developments in computer vision, such as 4D Gaussian Splatting, also show that learning the variation of Gaussian parameters over an additional dimension is an effective way to model complex dynamic structures. Similar latent conditioned GMM formulations have already been validated in cryo-EM by DynaMight. Our current work focuses on establishing a stable and scalable homogeneous backbone, and building a heterogeneous version by conditioning Gaussian parameters on a learned latent representation is a natural extension for our future work.

---

> > ### Comment · Reviewer_Mi5f · 2025-11-21
> >
> > Thanks for the authors’ clear response. After carefully reading the paper, supplementary materials, and all discussions, I feel comfortable accepting this work. I am also glad to see that the paper will be made public, as it will serve as a strong 3DGS-style baseline compared with prior methods.
> >
> > Meanwhile, I would like to note that a new paper titled “Reconstructing Heterogeneous Biomolecules via Hierarchical Gaussian Mixtures and Part Discovery” has been accepted to NeurIPS 2025. Its core idea is quite similar to CryoSplat and even supports heterogeneous reconstruction. I would suggest that the authors include a detailed discussion of this work in the related-works section, so that the unique contributions of CryoSplat can be further clarified.

---

> > > ### Author Response · Authors · 2025-11-22
> > > **Regarding “Reconstructing Heterogeneous Biomolecules via Hierarchical Gaussian Mixtures and Part Discovery” (CryoSPIRE)**
> > >
> > > We thank the reviewer for pointing out this very recent and excellent work. We will cite this paper and include additional discussion in the future version. We are pleased to see that several of its findings independently support and confirm statements made in cryoSplat. Specifically:
> > >
> > > 1. **Instability of previous GMM-based methods.**
> > >    The statement in Appendix C that *“the optimization dynamics are highly sensitive to the initial values”* supports our point in Line 131 that *“Without such guidance, random initialization leads to unstable optimization and poor reconstruction quality.”*
> > >    This reinforces that **GMM-based methods are widely considered challenging to train under random initialization**.
> > >
> > > 2. **Dependence on external map for initialization.**
> > >    The statement *“Given the input density map… Gaussians are then seeded at those positions.”* supports our point in Line 064 that *“They typically rely on consensus volumes from external pipelines, or even atomic models, for initialization…”*.
> > >    Although GMMs have been used in cryo-EM since E2GMM (2021), **this very recent NeurIPS 2025 work still relies on specialized initialization**, indicating that stability under random initialization has remained a challenging problem for GMM-based cryo-EM methods.
> > >
> > > 3. **Feasibility of heterogeneous reconstruction.**
> > >    The remark *“CryoSPIRE is in part inspired by Gaussian Splatting…”* and its great heterogeneous reconstruction results suggest that the design choices behind cryoSplat is **compatible with heterogeneous modeling as well**.
> > >
> > >
> > > We would like to emphasize our main contribution:
> > >
> > > **CryoSplat shows that GMMs can converge naturally and stably when the GMM-based image formation model is carefully aligned with cryo-EM imaging physics.**
> > >
> > > This highlights the strong potential of GMM-based representations in future cryo-EM analysis.
> > >
> > > Additionally, CryoSPIRE, consistent with prior GMM-based works, adopts **isotropic Gaussians**. Our implementation provides an efficient **anisotropic GMM renderer** that improves stability in optimization, which we believe can serve as a practical building block for future GMM-based approaches.
> > >
> > > Thanks again, and we will clarify the above points in the future revision.

---

### Official Review · Reviewer_xwVJ · 2025-10-31

**Soundness:** 4
**Presentation:** 3
**Contribution:** 3
**Rating:** 6
**Confidence:** 3

**Summary:**

The paper proposes cryoSplat, a new method for homogeneous reconstruction in cryo-electron microscopy (cryo-EM) using Gaussian mixture models (GMMs). cryoSplat leverages Gaussian splatting, a differentiable rendering technique, to achieve stable and efficient reconstruction directly from raw cryo-EM data. The authors introduce key innovations, including orthogonal projection-aware splatting and FFT-aligned coordinate systems, designed to improve the method's compatibility with cryo-EM's unique imaging physics. Experimental results on real datasets show that cryoSplat outperforms existing methods in terms of resolution, robustness to noise, and speed of convergence, providing a new foundation for GMM-based cryo-EM reconstruction.

**Strengths:**

### Innovative Methodology:

The integration of Gaussian splatting with cryogenic electron microscopy is novel and promising. This method represents an important step forward in making GMM-based approaches more feasible for cryo-EM.

The differentiable rendering approach combined with physically accurate projections offers a significant improvement over previous methods that relied on external priors or atomic models for initialization.

### Theoretical Soundness:

The authors carefully derive their projection model, which aligns well with the cryo-EM imaging process, addressing crucial issues like coordinate system alignment and view-dependent normalization. These innovations are necessary for physically accurate reconstructions and ensure that the method is well-grounded in cryo-EM physics.

### Empirical Validation:

The paper presents comprehensive experiments on real datasets, including datasets with varying levels of noise and structural complexity. The results show clear improvements in terms of resolution and robustness to heterogeneity.

The ablation studies and runtime efficiency analysis add further value, demonstrating the method’s ability to scale and deliver faster convergence compared to existing techniques like CryoDRGN.

### Practicality and Efficiency:

The CUDA-accelerated real-space renderer ensures that the method is computationally efficient, allowing for scalable optimization even with tens of thousands of Gaussians.

The use of simple but effective mean squared error (MSE) loss simplifies the optimization process, eliminating the need for complex regularizations while ensuring stable training.

**Weaknesses:**

### Limited Scope of Validation:

While the experiments on real datasets demonstrate strong performance, the method's robustness in heterogeneous reconstruction and in the face of ab initio reconstruction (with unknown particle poses) is not fully explored. The current work assumes known poses, which limits the scope of the method's applicability to real-world cryo-EM challenges.

Future work should explore how cryoSplat performs with pose estimation or heterogeneous reconstruction methods, as this is a critical challenge in cryo-EM.

### Dependency on Known Poses:

The current version of cryoSplat does not tackle the ab initio reconstruction problem where poses are unknown. It would be valuable to investigate how cryoSplat could be extended to tackle this challenge, which is a key strength of recent methods like CryoDRGN.

### Impact of Initialization:

While cryoSplat is robust in its random initialization, it would be useful to further analyze the potential impact of different initialization schemes. In certain cases, random initialization might still pose a challenge, especially for highly flexible or heterogeneous particles. Additional insights into initialization strategies could be helpful.

### Interpretability of Gaussian Mixtures:

The use of GMMs in cryo-EM reconstruction provides a compact and interpretable model, but more discussion on how this model relates to atomic-level resolution or finer structural details would be beneficial. For instance, how do the learned Gaussian parameters map to the structural features of macromolecules?

**Questions:**

### Handling Ab Initio Reconstruction:

While this paper shows strong results for homogeneous reconstruction with known particle poses, it would be insightful to know if the authors have any plans to extend cryoSplat to handle ab initio reconstruction. Specifically, how would the method perform when the particle orientations are unknown or the dataset is highly heterogeneous?

### Impact of Initialization:

The paper mentions that cryoSplat can converge from random initialization, but how sensitive is the method to the choice of initialization? Could the authors explore different initialization schemes, particularly in cases where the structure is highly flexible or there are ambiguities in the starting configurations?

### Model Interpretability:

GMMs are known for their interpretability, but in the context of cryo-EM, how do the learned Gaussian parameters (amplitude, mean, covariance) relate to the atomic-level structure? Could the authors provide more insights into how the reconstructed volumes align with the fine structural features of macromolecules?

### Performance in Heterogeneous Systems:

The current paper focuses on homogeneous reconstruction. How would cryoSplat perform in heterogeneous reconstruction scenarios where different conformations coexist in the same dataset? Could the method be extended or adapted for more complex, heterogeneous systems?

**Details Of Ethics Concerns:**

No Ethics Concerns

---

> ### Author Response · Authors · 2025-11-21
>
> ### **1. Limited Scope**
>
> Typically, GMM-based methods are regarded unstable, which require special initializations or constraints to keep their stability during training, as discussed in Sec. 2.1.3. 3DGS used in novel view synthesis also require initialization from given point clouds. We believe the main contribution of our paper is to show that, by carefully tuning Gaussian splatting technique with the cryo-EM imaging procedure, a GMM can naturally and stably converge. In the meantime, the experiments also show that GMMs keep great consistency while converging on extremely noisy data, which is a promising property to be used in ab initio reconstruction. As discussed in Sec. 5, we are indeed working on ab initio reconstruction and seeing some promising results. However, that task has its own challenges. For example, how to choose a proper strategy for GMMs to control the range and interval of pose search is still an open question. The whole ab initio pipeline still needs to be further tuned.
>
> ### **2. Impact of Initialization**
>
> Thank you for the question regarding initialization sensitivity. As shown in Sec. E.9, our method is highly robust to both stochastic and structural variations in initialization. Varying the random seed (0–9) produces nearly identical FSC curves, and changing the spatial spread of initial Gaussian placements still leads to essentially the same final reconstruction. These results confirm that our approach does not rely on carefully designed initialization and consistently avoids unfavorable local minima.
>
> ### **3. Model Interpretability**
>
> Compared with other representations, a promising attribute of a GMM is that it is separable in real space. In other words, each Gaussian can move independently. Thus, a subset of Gaussians can naturally correlate to a functional group of a macrobiomolecule. For example, DynaMight uses GMMs to interpret the deformation of biomolecules with conformational heterogeneity. CryoSTAR uses GMMs as bridges between atomic models and density maps, applying atom-related constraints. These previous works have been discussed and summarized in Sec. 2.1.3. In summary, the position of Gaussians correlates positively with the atoms of biomolecules. Note that there is no solid evidence showing that each Gaussian will finally link to each atom of a molecule, but it is a promising direction to be further explored. Besides, as the DC component has been mostly filtered out by CTF during cryo-EM imaging, the amplitudes of Gaussians will not be the absolute value of electrostatic potential. However, we believe they have a roughly linear relationship.
>
> ### **4. Heterogeneous Reconstruction**
>
> For heterogeneous reconstruction, many other studies have applied GMMs to it, as discussed in Sec. 2.1.3. A common paradigm is to add a VAE before a Gaussian renderer, which takes a particle-related input and outputs Gaussian parameters. As is discussed, a common issue of previous methods is that their GMMs fail to naturally and stably converge. The ablation study in Appendix E.3 shows that a potential factor contributing to the isotropic GMMs is prone to salient local minima, while anisotropic GMM renderer is harder to build. We fixed this issue by carefully adapting the Gaussian splatting technique to cryo-EM imaging. The experiments in this paper show that the more Gaussians, the better performance, and we offer a stable and efficient implementation which can be used as a building block for future methods. As discussed in Sec. 5, we decide to leave this direction for future work.

---

### Author Response · Authors · 2025-11-21

We thank all reviewers for their time and constructive feedback. The comments have helped us clarify the methodological contributions, refine the experimental analysis, and improve the overall presentation. We are encouraged to see recognition of the technical depth and the novelty of integrating Gaussian splatting with cryo-EM reconstruction, and we have revised the manuscript to elaborate these aspects more clearly. To facilitate verification, we have provided an anonymous demo in the supplementary material that reproduces the main results, and the full codebase will be released upon publication. We address each reviewer’s comments individually.

---

### Meta-Review · Area_Chair_ikUS · 2026-01-25

**Summary:**

(6,6,6,2) This paper presents CryoSplat, a Gaussian splatting method for homogeneous cryo-EM reconstruction, assuming known poses. While the scope of the work is a simplified setting for cryo-EM (one reviewer notes it as already "solved"), the works provides a well-executed 3DGS-style baseline for cryo-EM homogeneous reconstruction. There were some concerns about the baselines and comparison to standard tools that were addressed during rebuttals.

**Reviewer Concerns:**

Reviewers qU3z and Mi5f both raised the concern that CryoSplat density maps appear smoother and in some cases miss fine details that voxel and neural field methods recover, despite higher FSC. The authors diagnose this as a known limitation of half-map FSC with biased but self-consistent representations such as GMMs.

The authors supply an anonymized demo implementation in the rebuttal that reproduces main results and commit to releasing the full code upon publication, which directly addresses the reproducibility concern.

**Reviewer Scores:**

N/A

---

### Decision · Program_Chairs · 2026-01-26

Accept (Poster)